# Purest Quantum State Identification

**Yingqi Yu[1]  Honglin Chen[1]  Jun Wu[1]  Wei Xie[1]\*  Xiang-Yang Li[1,2]†**

[1]School of Computer Science and Technology, University of Science and Technology of China
[2]Hefei National Laboratory, University of Science and Technology of China
{yingqiyu, chl777, devilu}@mail.ustc.edu.cn
{xxieww, xiangyangli}@ustc.edu.cn

## Abstract

Quantum noise constitutes a fundamental obstacle to realizing practical quantum technologies. To address the pivotal challenge of identifying quantum systems least affected by noise, we introduce the purest quantum state identification, which can be used to improve the accuracy of quantum computation and communication. We formulate a rigorous paradigm for identifying the purest quantum state among $K$ unknown $n$-qubit quantum states using total $N$ quantum state copies. For incoherent strategies, we derive the first adaptive algorithm achieving error probability $\exp\left(-\Omega\left(\frac{NH_1}{\log(K)2^n}\right)\right)$, fundamentally improving quantum property learning through measurement optimization. By developing a coherent measurement protocol with error bound $\exp\left(-\Omega\left(\frac{NH_2}{\log(K)}\right)\right)$, we demonstrate a significant separation from incoherent strategies, formally quantifying the power of quantum memory and coherent measurement. Furthermore, we establish a lower bound by demonstrating that all strategies with fixed two-outcome incoherent POVM must suffer error probability exceeding $\exp\left(-O\left(\frac{NH_1}{2^n}\right)\right)$. This research advances the characterization of quantum noise through efficient learning frameworks. Our results establish theoretical foundations for noise-adaptive quantum property learning while delivering practical protocols for enhancing the reliability of quantum hardware.

## 1 Introduction

Quantum computing demonstrates the remarkable potential for solving computational problems that are intractable for classical systems [1, 2]. However, current quantum systems face two fundamental constraints that limit their practical applications: (1) the number of available quantum bits (qubits) in existing devices remains severely limited compared to theoretical requirements [3, 4], and (2) these qubits exhibit high sensitivity to environmental noise, compromising the reliability of quantum operations [5]. These fundamental constraints impose substantial barriers to achieving reliable quantum control and measurement, prerequisites for unlocking practical quantum computational advantages. Owing to variations in device implementation and environmental factors, the performance of quantum computation and communication differs markedly among various quantum systems [6]. Thus, identifying the optimal quantum device [5], quantum state, or quantum channel [7] is significant for the near-term quantum science and technology.

To evaluate the quality of quantum systems, researchers often use the purity of quantum states as a criterion. In quantum communication, high-accuracy quantum channels are critical to ensuring the accurate transmission of quantum information [7]. By measuring the purity of the channel's

---

\*Corresponding author
†Corresponding author

39th Conference on Neural Information Processing Systems (NeurIPS 2025).

associated state (Choi-state [7]), researchers can evaluate how noise affects the channel. On the other hand, preparing high-purity quantum states is also the foundation for quantum algorithms. The majority of quantum algorithms rely on pure quantum states as inputs [1, 8, 9]. Therefore, identifying high-purity quantum states is essential for accurate quantum computing.

In this paper, we study the problem of purest quantum state identification (PQSI). The learner aims to identify the purest state among multiple unknown quantum states using $N$ copies across all states. This issue holds considerable importance from both theoretical and practical perspectives. By identifying the purest quantum state, it is achievable to distinguish the best quantum system that is least affected by noise, thus improving the accuracy of quantum computing and quantum communication in near-term quantum systems.

This paper makes the following key contributions:

**Problem Model.** To the best of our knowledge, this is the first study dedicated to the best quantum state identification. In this paper, we focus on the issue of the purest quantum state identification, i.e., identifying the purest quantum state from a set of available quantum states. In this problem, while allocating copies to different quantum systems, the learner also needs to select the basis for quantum measurement, which significantly increases its complexity. Due to the limited number of qubits in existing quantum devices, these systems may not be able to facilitate large-scale quantum measurement [10, 11]. Consequently, while coherent (multi-copy) measurement techniques can more efficiently acquire information about quantum states, research on incoherent (single-copy) measurement is more practical. We formalize the problem of purest quantum state identification with incoherent (single-copy) measurement as follows:

**Problem 1.1** (Purest quantum state identification (PQSI) with incoherent (single-copy) measurement)**.** *There is a set of $K$ unknown $n$-qubit quantum states represented as $S = \{\rho_1, \ldots, \rho_K\}$. The learner aims to identify the purest quantum state in $S$ using a total of $N$ copies across all states. The problem protocol at round $t \in \{1, ..., N\}$ is as follows:*

- *The learner chooses a quantum state $\sigma_t$ from the set $S$ and gets a copy of it.*

- *The learner selects a POVM and uses it to measure $\sigma_t$, after which $\sigma_t$ is destroyed.*

*Upon completing $N$ measurements, the learner selects a quantum state $\rho' \in S$ based on the measurement outcomes. The objective of the learner is to maximize the probability of identifying the purest quantum state, i.e., $\mathbb{P}(\rho' \in \arg\max_{\rho \in S} \mathrm{Tr}(\rho^2))$.*

In general, when coherent measurements are available, we formalize the problem as follows:

**Problem 1.2** (Purest quantum state identification(PQSI) with coherent (multi-copy) measurement)**.** *There is a set of $K$ unknown $n$-qubit quantum states represented as $S = \{\rho_1, \ldots, \rho_K\}$. The learner aims to identify the purest quantum state in $S$ using a total of $N$ copies across all states. Let $N_t$ denote the number of quantum state copies that remain available in round $t$ and $N_1 = N$. When $N_t > 0$, the problem protocol at round $t \in \mathbb{N}^+$ is as follows:*

- *The learner decides the number of copies of the quantum state $\rho_i$ to acquire, denoted as $s(i, t)$, where $s(i, t) \geq 0$ and $1 \leq \sum_{i=1}^{K} s(i, t) \leq N_t$. The quantum state constructed from these copies can be represented as $\sigma_t = \rho_1^{\otimes s(1,t)} \otimes \ldots \otimes \rho_K^{\otimes s(K,t)}$. Let $N_{t+1} = N_t - \sum_{i=1}^{K} s(i, t)$.*

- *The learner selects an entangled POVM and uses it to measure $\sigma_t$, after which $\sigma_t$ is destroyed.*

*Upon completing measurements of all $N$ copies of quantum states, the learner selects a quantum state $\rho' \in S$ based on the measurement outcomes. The objective of the learner is to maximize the probability of identifying the purest quantum state $\mathbb{P}(\rho' \in \arg\max_{\rho \in S} \mathrm{Tr}(\rho^2))$.*

**Algorithms.** We develop two distinct algorithms to address this problem in different settings. To simplify the expression, for $i \in \{1, ..., K\}$, let $\rho_{(i)}$ be the $i$-th purest quantum state in $S$ and $\Delta_{(i)} = \mathrm{Tr}\left(\rho_{(1)}^2\right) - \mathrm{Tr}\left(\rho_{(i)}^2\right)$ be the purity gap between the optimal quantum state and the $i$-th purest quantum state. In scenarios where only single-copy measurements are available, we developed the incoherent measurement based algorithm IM-PQSI. We employ Haar unitary matrices to generate random measurement basis and to measure quantum states in $S$. This measurement approach allows us to evaluate the purity of quantum states by analyzing measurement expectations and variances. The Haar unitary matrices utilized in IM-PQSI belong to the class of unitary 4-designs,

which can be efficiently simulated on a quantum computer up to inverse-exponential trace distance [12, 13, 14, 15, 16]. During the evaluation of quantum state purity, our algorithms dynamically allocate copies to various quantum states. By assigning more copies to states with higher purity, we enhance the estimation accuracy of these states, thereby increasing the likelihood of identifying the purest quantum state. Furthermore, we improve the performance of the algorithm for identifying the optimal quantum state by balancing the selection of measurement basis with the number of measurements conducted for each base. The error probability of this algorithm satisfies the following theorem:

**Theorem 1.3** (Informal version of Theorem 4.1). *There exists an algorithm that solves the problem of the purest quantum state identification with incoherent measurement whose error probability satisfies*

$$e_N \leq \exp\left(-\Omega\left(\frac{NH_1}{\log(K)2^n}\right)\right),\tag{1}$$

*where $H_1 = \min_{i \in \{2,...,K\}} \frac{\Delta_{(i)}}{i}$.*

Conversely, when multi-copy measurements are accessible, we developed the coherent measurement based algorithm CM-PQSI. We use the SWAP test to estimate the purity of quantum states in this algorithm, and its error probability satisfies the following theorem:

**Theorem 1.4** (Informal version of Theorem 6.1). *There exists an algorithm that solves the problem of the purest quantum state identification with coherent measurement whose error probability satisfies*

$$e_N \leq \exp\left(-\Omega\left(\frac{NH_2}{\log(K)}\right)\right),\tag{2}$$

*where $H_2 = \min_{i \in \{2,...,K\}} \frac{\Delta_{(i)}^2}{i}$.*

By comparing the error probability upper bound of these two algorithms, we can identify the advantages of quantum memory.

**Lower Bound.** For incoherent measurements, we utilize the properties of Haar unitary matrices to formulate a related problem named Purest Random Quantum State Identification (PRQSI). In this context, the learner is required to consider a specific quantum state distribution constructed from Haar unitary matrices. We demonstrate that the lower bound of the error probability for the PRQSI problem is also applicable as the lower bound for the error probability of the PQSI problem. Furthermore, we show that the measurement outcomes generated by three-fourths of the Haar unitary matrices are insufficient for the learner to easily differentiate among them, presenting a challenge in identification. Additionally, we demonstrate that in any PRQSI problem employing a fixed two-outcome Positive Operator-Valued Measure, the measurement outcomes exhibit a Bernoulli distribution that is inherently difficult to distinguish. Consequently, we derive the lower bounds for the error probability of the PRQSI, which are also the lower bounds for the error probability of the PQSI problem, as follows:

**Theorem 1.5** (Informal version of Theorem 5.6). *For any algorithm $\mathcal{A}$ to solve the purest quantum state identification using fixed 2-outcome randomly incoherent POVM, there exists a set of quantum states which makes the error probability of $\mathcal{A}$ satisfies*

$$e_N \geq \exp\left(-O\left(\frac{NH_1}{2^n}\right)\right),\tag{3}$$

*where $H_1 = \min_{i \in \{2,...,K\}} \frac{\Delta_{(i)}}{i}$.*

## 2 Related Work

**Quantum learning and testing.** Quantum learning and testing [17, 18] is a vital area of research in quantum computing and quantum communication. Extensive investigations have been conducted to understand the complexities of various measurements. Quantum state tomography [19, 20, 21, 22] involves obtaining complete information about the density matrix of a quantum state through measurements. While this technique can be employed to tackle the PQSI problem, it incurs significant sampling costs. For the quantum state certification [23, 24, 25, 26], the target is to determine whether

a quantum state is close to a specific target quantum state. Our problem can be viewed as identifying the quantum state that is the farthest from the maximally mixed state. However, this problem is focused on quantum testing and does not deal with distance estimation.

Hence, these methods are not applicable to the PQSI problem. Another category of problems relates to inner product estimation between two quantum states [27, 28, 29, 30]. When proving lower bounds, this category often significantly restricts the quantum states for distinction. The relevant literature employs two general approaches to establish problem complexity. The first approach involves constructing counterexamples using Haar unitary matrices [27, 22, 26, 23]. However, the representation-theoretic structure of Haar unitary matrices is intricate [31], which makes it difficult to use. The other approach uses Gaussian Orthogonal Ensemble matrices to create counterexamples [24, 21]. However, when using GOE to prove the lower bound, the distance between quantum states is in a specific range rather than a fixed number, making it unsuitable for the PQSI problem.

**Classical best arm identification.**   To the best of our knowledge, our work is the first to consider the best quantum state identification. Among the classical learning tasks, the best arm identification [32, 33, 34, 35, 36] has been extensively studied and is divided into two categories: fixed budget [37] and fixed confidence [38]. However, the existing research can only deal with the problem under specific distributions. These limitations restrict the algorithm's applicability and leave considerable room for further research on this issue. In our problem, we must select an appropriate POVM basis while choosing the quantum state in each round. The quantum state space and the POVM space grow exponentially with the increase in qubits, which makes this problem significantly more challenging than solving a classical problem of the best arm identification.

## 3   Preliminaries and Notations

**Quantum State.**   Let $d = 2^n$ denote the dimension of an $n$-qubit quantum system. An $n$-qubit quantum state can be represented by a density matrix $\rho \in \mathbb{C}^{d \times d}$, which is Hermitian and trace-1 positive semi-definite. In particular, an $n$-qubit pure quantum state can be represented by a unit vector $|\psi\rangle \in \mathbb{C}^d$. The purity of a quantum state $\rho$ is $\text{Tr}(\rho^2)$.

**Quantum Measurement.**   Quantum measurements are usually described by a Positive Operator-Valued Measure (POVM), which produces probabilistic outcomes. An $n$-qubit positive operator-valued measurement $\mathcal{M}$ can be represented as a collection of positive semi-definite matrices $\mathcal{M} = \{M_m\}_m$, where $M_m \in \mathbb{C}^{d \times d}$ and $\sum_m M_m = I_d$. When using $\mathcal{M}$ to measure a quantum state $\rho$, the probability of outcome $m$ is $\text{Tr}(M_m \rho)$, and the quantum state $\rho$ is destroyed. When the coherent measurement method is employed, the learner can perform entanglement measurements on quantum states $\rho_1 \otimes \ldots \otimes \rho_m$. However, this approach necessitates the support of large-scale quantum devices and quantum memory, which are not feasible with current quantum technologies. Therefore, researching incoherent measurement methods applicable to near-term quantum devices is of great significance.

In this study, we aim to identify the purest quantum state from a set of unknown quantum states, achieving the highest probability through $N$ measurements. For the purpose of simplicity, we will assume that there exists a unique optimal quantum state that is the purest in the set $S$, denoted as $\mu^\star = \mu_{i^\star}$. For $i \neq i^\star$, we represent the purity difference between each non-optimal quantum state and the optimal quantum state using

$$\Delta_i = \text{Tr}\left(\rho_{i^\star}^2\right) - \text{Tr}\left(\rho_i^2\right).$$

For $i \in \{1, ..., K\}$, let $\rho_{(i)}$ be the $i$-th purest quantum state in $S$, then we have

$$\text{Tr}\left(\rho_{i^\star}^2\right) = \text{Tr}\left(\rho_{(1)}^2\right) > \text{Tr}\left(\rho_{(2)}^2\right) \geq ... \geq \text{Tr}\left(\rho_{(K)}^2\right),$$

and

$$\Delta_{(2)} \leq \Delta_{(3)} \leq ... \leq \Delta_{(K)}.$$

Let $e_N$ denote the probability that the learner does not choose the purest quantum state in $S$ after $N$ samples and measurements, i.e., $e_N = \mathbb{P}\left(\rho' \notin \arg\max_\rho \text{Tr}(\rho^2)\right)$. The learner's objective is to $\min e_N$. We summarize key notations used throughout this paper in Table 1.

# 4 Algorithm for PQSI with incoherent measurement

The current era of quantum computing presents limitations in the number of available qubits. This may hinder the measurement of multiple quantum states operated jointly. In this section, we use incoherent (single-copy) measurement methods in each round to select the purest quantum state from the set of quantum states.

The algorithm we designed to solve the PQSI problem with incoherent measurements is shown in Algorithm 1. In our algorithm, we utilize Haar-random unitary matrices to construct multiple random measurement bases for probing quantum state copies. For each constructed measurement basis, we perform $m$ repeated measurements on the same quantum state. Let the $i$-th measurement outcome be $x_i \in \{0, \ldots, d-1\}$ for a $d$-outcome POVM. We estimate the quantum state's purity based on the empirical probability of obtaining identical outcomes, defined as

$$\tilde{g} = \frac{1}{m(m-1)} \sum_{i=1}^{m} \sum_{j \neq i} \mathbb{1}\{x_i = x_j\},$$

which corresponds to the well-known collision estimator [39, 40] used to estimate the second moment $\sum_{i=1}^{d} p_i^2$ in classical statistics.

The rationale behind this choice is as follows. The positive operator-valued measurements (POVMs) adopted in our algorithm have $d$ possible outcomes, where $p_i$ denotes the probability of observing the $i$-th outcome. The purity of a quantum state is closely related to the quantity $\sum_{i=1}^{d} p_i^2$. Hence, an essential step in our algorithm is to estimate this second-order statistic accurately. The collision estimator provides a natural and statistically efficient approach to this task, allowing us to connect the observed measurement coincidences to the underlying purity of the quantum state. By analyzing the expectation and variance of $\tilde{g}$, we can further quantify the probability of correctly identifying the purest quantum state within our framework.

Algorithm 1 includes two random processes. The first process involves the random selection of Haar unitary matrices to construct measurement bases, while the second process entails the random acquisition of measurement results using these bases. Balancing the estimation errors introduced by these two processes is crucial for enhancing the algorithm's accuracy. When the purity difference between quantum states is substantial, each measurement base requires fewer instances to distinguish purity differences among states effectively. Conversely, when the purity difference is minimal, more measurements are necessary under a single measurement base to gather sufficient information, limiting the number of measurement bases that can be utilized. Furthermore, when the purity difference is extremely slight, the choice of measurement basis significantly influences the measurement outcomes. Performing $\Theta(d^2)$ measurements under a single measurement base is essential for adequately differentiating the outcomes between quantum states. Additionally, increasing the number of measurements under one measurement base will not improve the accuracy of purity estimation. Given these considerations, we balance the number of measurement bases and the number of measurements per base to enhance the probability of identifying the optimal quantum state.

In Algorithm 1, we use Haar unitary matrices to construct random measurement bases. Haar unitary matrices are extensively employed in quantum property testing and learning theory [41, 27, 30]. Achieving exact Haar randomness is challenging; therefore, unitary $t$-designs are typically used to approximate Haar unitary matrices. Haar unitary matrices used in Algorithm 1 can be approximated using unitary 4-designs. As established in [12], unitary $t$-designs can be efficiently implemented on quantum computers with exponentially minor approximation errors. This theoretical foundation ensures the practical feasibility and mathematical validity of employing Haar unitary matrices in our algorithmic framework.

Then, we give the error probability upper bound of IM-PQSI in the following theorem.

**Theorem 4.1.** *For $i \in \{1, ..., K\}, \Delta_i \geq c > \frac{1}{d^2}$, where $c$ is a constant. Set $m = \lceil \frac{1}{\sqrt{c}} \rceil$ in Algorithm 1. The error probability of Algorithm 1 satisfies*

$$e_N \leq \frac{K(K-1)}{2} \exp\left(-\Omega\left(\frac{\sqrt{c}NH_1}{\overline{\log}(K)d}\right)\right), \tag{4}$$

*where $H_1 = \min_{i \in \{2, ..., K\}} \frac{\Delta_{(i)}}{i}$.*

---

**Algorithm 1** Incoherent measurement based algorithm for solving PQSI problem (IM-PQSI)

---

**Input:** Copy access to $S = \{\rho_1, ..., \rho_K\}$, sample number $N$.

**Initialization:** Set $S_0 = \{\rho_1, ..., \rho_K\}$, $\overline{\log}(K) = \frac{1}{2} + \sum_{i=2}^{K} \frac{1}{i}$, $N_0 = 0$ and $N_k = \left\lceil \frac{1}{\overline{\log}(K)} \frac{N-K}{K+1-k} \right\rceil$, for $k \in \{1, ..., K-1\}$. Sample $\lfloor N/m \rfloor$ random unitary matrix $U_1, ..., U_{\lfloor N/m \rfloor}$ according to the Haar measure.

**for** k=1,..., K-1 **do**

    **for** $\rho \in S_{k-1}$ and $j \in \{\lfloor \frac{N_{k-1}}{m} \rfloor + 1, ..., \lfloor \frac{N_k}{m} \rfloor\}$ **do**

        Measure $m$ copies of $\rho$ in the basis $\{U_j^\dagger |i\rangle\langle i| U_j\}_{i=0}^{d-1}$ and set the outputs as $x(\rho, j, 1), ..., x(\rho, j, m)$.

        Let $\tilde{g}(\rho, j) = \frac{1}{m^2} \sum_{i=0}^{d-1} \left[ \sum_{l=1}^{m} \mathbb{1}\{x(\rho, j, l) = i\} \right]^2 - \frac{1}{m}$.

    **end for**

    Let $w(\rho, k) = \frac{1}{\lfloor \frac{N_k}{m} \rfloor} \sum_{j=1}^{\lfloor \frac{N_k}{m} \rfloor} \tilde{g}(\rho, j)$.

    Let $S_k = S_{k-1} \setminus \arg\min_{\rho \in S_{k-1}} w(\rho, k)$.

**end for**

Output the quantum state $\rho'$ in $S_{k-1}$.

---

*Proof Sketch.* The expectation of $w(\rho, k)$ and $\tilde{g}_{\rho, j}$ in Algorithm 1 satisfies $\mathbb{E}[w(\rho, k)] = \mathbb{E}[\tilde{g}_{\rho, j}] = \frac{(m-1)(1+\mathrm{Tr}(\rho^2))}{m(d+1)}$, and the variance of $\tilde{g}(\rho, j)$ satisfies $\mathrm{Var}(\tilde{g}(\rho, j)) = O\left(\frac{1}{d^3} + \frac{1}{m^2 d} + \frac{1}{md^2}\right)$. By Bernstein's inequality and the union bound of error probability, we have $e_n \leq \frac{K(K-1)}{2} \exp\left(-\Omega\left(\frac{\sqrt{c}NH_1}{\overline{\log}(K)d}\right)\right)$ where $H_1 = \min_{i \in \{1, ..., K\}} \frac{\Delta_{(i)}}{i}$. The proof details are provided in Appendix B.2. $\qquad\qquad\qquad\square$

When the purity gaps between quantum states are very small, the gap parameter $H_1$ becomes correspondingly small, which naturally results in a looser error bound. This reflects the intrinsic difficulty of the problem: identifying the quantum state with the highest purity is fundamentally hard when multiple states have nearly identical purities. Moreover, the system dimension $d = 2^n$ grows exponentially with the number of qubits $n$, so that $\frac{1}{d^2}$ tends to zero as $n$ increases. Therefore, in Theorem 4.1, we assume that for any quantum state $\rho \in S_\rho$, $\mathrm{Tr}(\rho^{\star 2}) - \mathrm{Tr}(\rho^2) \geq c > \frac{1}{d^2}$. If this assumption does not hold, we can instead derive the following conclusion:

**Lemma 4.2.** *Set $m = d$ in Algorithm 1. The probability of error of Algorithm 1 satisfies*

$$e_N \leq \frac{K(K-1)}{2} \exp\left(-\Omega\left(\min\left(\frac{NH_2}{\overline{\log}(K)}, \frac{NH_1}{\overline{\log}(K)d^2}\right)\right)\right), \tag{5}$$

*where $H_1 = \min_{i \in \{2, ..., K\}} \frac{\Delta_{(i)}}{i}$, and $H_2 = \min_{i \in \{2, ..., K\}} \frac{\Delta_{(i)}^2}{i}$.*

Appendix B.3 provides the proof details of Lemma 4.2.

In the field of quantum learning and testing, research on quantum channels constitutes a critical aspect. Evaluating the impact of noise on quantum channels can significantly enhance the accuracy of quantum computing and quantum communication [7]. Introduced a method for assessing the "unitarity" of a quantum channel by evaluating the purity of a quantum state. Subsequently, we can utilize the algorithm IM-PQSI to identify the most "unitary" quantum channel from a quantum channel set. Let $u_{(i)}$ denote the unitarity of the $i$-th most unitary quantum channel. We have the following corollary:

**Corollary 4.3.** *There exists an algorithm that solves the problem of the most "unitary" channel identification with incoherent access whose error probability satisfies:*

$$e_N \leq \exp\left(-\Omega\left(\frac{NH_u}{\overline{\log}(K)2^n}\right)\right), \tag{6}$$

*where $H_u = \min_{i \in \{2, ..., K\}} \frac{u_{(1)} - u_{(i)}}{i}$.*

# 5 Lower bound for PQSI with incoherent measurement

In this section, we investigate the lower bound on the error probability for solving the problem of purest quantum state identification. This problem requires distinguishing between quantum states with different purity through sampling and measurement. Recent studies [31] indicated that when a quantum state $\rho$ is rotated by a Haar unitary matrix and measured $N$ times, the output distribution can be calculated only if $\rho$ is either a pure state or a maximally mixed state. Consequently, the complexity analysis of testing problems often assumes that one of the quantum states is either a pure state or a maximally mixed state. This limitation presents significant challenges to our analysis.

To solve this problem, we reduce the problem of identifying the purest quantum state to the problem of identifying the purest random quantum state. This reduction allows us to retain the problem's complexity while enabling us to analyze the complexity by considering only a single problem instance.

Next, we demonstrate that for any POVM base $\mathcal{M}$, there is a set of unitary matrices $\mathbb{U}(\mathcal{M})$ satisfying that (1) $\mathbb{P}_{U \sim \text{Haar}}(U \in \mathbb{U}(\mathcal{M})) = \Omega(1)$; and (2) when the quantum states rotated by these unitary matrices, they are difficult to distinguish by the POVM base $\mathcal{M}$.

At last, we only consider all the possible POVM $\mathcal{M}$ and their corresponding set of unitary matrix $\mathbb{U}(M)$. By analyzing the sampling distribution for specific POVM $\mathcal{M}$ and unitary matrix in $\mathbb{U}(M)$, we reduce the problem into a classical problem for resolution and provide a lower bound for the purest quantum state identification.

Similar to Definition 7 in [42], we analyze the lower bound of the error probability for any algorithm solving the purest quantum state identification problem using a 2-outcome randomly incoherent POVM to evaluate the task's difficulty.

**Definition 5.1** (Randomly fixed incoherent two-outcome POVM). We say an algorithm $\mathcal{A}$ with a randomly fixed incoherent two-outcome POVM, if it proceeds as the following: The algorithm $\mathcal{A}$ samples a POVM $\mathcal{M} = \{M_0, M_1 = I_d - M_0\}$ from a well-designed distribution of POVMs $\mathcal{D}_{\mathcal{M}}$ and performs the two-outcome single-copy POVM $\mathcal{M}$ on the copies of the quantum states.

## 5.1 Problem reduction

In this subsection, we aim to demonstrate that if there exists a set of random quantum states $T(x) = \{\tau_1(x), \ldots, \tau_K(x)\}$ which is difficult to identify the purest one in $T$, there also exists a corresponding set of quantum states $S = \{\rho_1, \ldots, \rho_K\}$ where is difficult to identify the purest one in $S$. In this way, we only need to construct $K$ random quantum states, which are hard to distinguish. Then, we can demonstrate the difficulty of the purest quantum state identification problem.

To enhance our discussion, we define the problem of the purest random quantum state identification(PRQSI) with incoherent measurement as follows:

**Problem 5.2** (Purest random quantum state identification(PRQSI) with incoherent measurement). *Consider a set of $K$ unknown random quantum states, denote as $T(U) = \{\tau_1(U), \ldots, \tau_K(U)\}$, where $U$ samples from a fixed distribution $\mathcal{D}$. For each $k \in [K]$ and $U \sim \mathcal{D}$, $\text{Tr}((\tau_k(U))^2) = z_k$. In each round $t \in \{1, ..., N\}$, the learner selects an index $k_t \in [K]$ and a POVM $\mathcal{M}_t$. The learner obtains a copy of $\tau_{k_t}(U)$ and uses $\mathcal{M}_t$ to measure it. Upon completing $N$ measurements, the learner selects an index $k' \in [K]$ as the output. The objective of the learner is to maximize $\mathbb{P}(k' \in \arg\max_{k \in [K]} z_k)$.*

As shown in the following lemma, we can reduce the proof of the error probability lower bound for the PQSI problem into the proof of the lower bound for a specific instance of the PRQSI problem.

**Lemma 5.3.** *If there exists a set of random quantum states $T(U)$ in Problem 5.2 such that any algorithm $\mathcal{A}_T$ addressing Problem 5.2 cannot identify the purest random quantum state with an error probability lower than $e_N$, then for any algorithm $\mathcal{A}$ addressing Problem 1.1, there exists a specific set of quantum states $S = \{\rho_1, \ldots, \rho_K\}$ such that the error probability of algorithm $\mathcal{A}$ is not lower than $e_N$.*

*Proof Sketch.* Suppose that there exists an algorithm $\mathcal{A}$ satisfying whose error probability for solving Problem 1.1 is less than $e_N$. We can prove that the algorithm $\mathcal{A}$ can solve the Problem 5.2 with

the error probability less than $e_N$. Furthermore, we can establish the proof by considering the contrapositive of this statement. Detailed explanations of the proof are included in Appendix C.1. □

According to Lemma 5.3, we will establish the lower bound of the error probability for the problem PQSI by demonstrating the error probability lower bound for the following problem:

**Problem 5.4.** *Consider the Problem 5.2. For* $k \in [K]$, *let* $\alpha_k = \sqrt{\frac{dz_k - 1}{d - 1}}$, *and* $\tau_k(U) = \alpha_k U |0\rangle\langle 0| U^\dagger + \frac{1 - \alpha_k}{d} I_d$, *where* $U \sim \text{Haar}$.

Then, In the Problem 5.4, for $k \in \{1, ..., K\}$ and $U \sim \text{Haar}$, the purity of the quantum state $\tau_k(U)$ satisfies:

$$\text{Tr}\left((\tau_k(U))^2\right) = \left(\frac{1 + (d-1)\alpha_k}{d}\right)^2 + (d-1)\left(\frac{1 - \alpha_k}{d}\right)^2 = z_k.$$

## 5.2 Random quantum state purity certification

To analyze Problem 5.4, we first study the properties of the measurement results obtained from conducting $N'$ measurements on the sampled quantum states from the quantum state distribution $\mathcal{D} = \{\rho | \rho = \alpha U |0\rangle\langle 0| U^\dagger + \frac{1-\alpha}{d} I_d\}$, using a specific POVM $\mathcal{M} = \{M_0, M_1\}$, where $U \sim \text{Haar}$ and $\alpha$ is a constant satisfying $0 \le \alpha \le 1$.

**Lemma 5.5.** *Let* $a \in [0, 1]$. *Using a specific POVM* $\mathcal{M} = \{M_0, M_1\}$ *to measure the random quantum state in* $\mathcal{D} = \{\rho | \rho = \alpha U |0\rangle\langle 0| U^\dagger + \frac{1-\alpha}{d} I_d\}$. *Let* $M = \arg\min_{M' \in \{M_0, M_1\}} \text{Tr}(M')$. *We have*

$$\mathbb{P}_{U \sim \text{Haar}} \left[ \left| p_{\mathcal{M}}(M|U) - \frac{\text{Tr}(M)}{d} \right| < \frac{2a\sqrt{\text{Tr}(M)}}{d} \right] \ge \frac{3}{4},$$

*and there is a function* $c(\mathcal{M}, U)$ *satisfying* $p_{\mathcal{M}}(M|U) - \frac{\text{Tr}(M)}{d} = c(\mathcal{M}, U)\alpha$.

*Proof Sketch.* By utilizing the properties of the Haar unitary matrix, we can calculate the variance of $p_{\mathcal{M}}(M|U)$ and prove the probabilistic bounds in the lemma using Chebyshev's inequality, thus completing the proof. The proof details are provided in Appendix C.2. □

According to Lemma 5.5, for a specific POVM $\mathcal{M}$ and unitary matrix $U$, let $\mathbb{U}_{\mathcal{M}}$ denote the set of unitary matrix satisfying that $\mathbb{U}_{\mathcal{M}} = \left\{ U : \left| p_{\mathcal{M}}(M|U) - \frac{\text{Tr}(M)}{d} \right| < \frac{2\alpha\sqrt{\text{Tr}(M)}}{d} \right\}$. We have $\mathbb{P}_{U \sim \text{Haar}}(U \in \mathbb{U}_{\mathcal{M}}) \ge \frac{3}{4}$. The following analysis will focus on the unitary matrices in $\mathbb{U}_{\mathcal{M}}$.

## 5.3 Error probability lower bound

In this subsection, we will prove the lower bound of error probability for using algorithms to solve Problem 1.1 and Problem 5.4.

Let $M = \arg\min_{M' \in \{M_0, M_1\}} \text{Tr}(M')$, then we have $\text{Tr}(M) \in [0, d/2]$. In the following theorem, let $\text{Tr}(M) > 16$ in order to make $\text{Tr}(M) - 2\sqrt{\text{Tr}(M)} \ge \frac{1}{2}\text{Tr}(M)$.

**Theorem 5.6.** *Let* $M = \arg\min_{M' \in \{M_0, M_1\}} \text{Tr}(M')$ *and* $\text{Tr}(M) > 16$. *For any algorithm* $\mathcal{A}$ *to solve the purest quantum state identification using fixed 2-outcome randomly incoherent POVM, there exists a set of quantum states which makes the error probability of* $\mathcal{A}$ *satisfies*

$$e_N \ge \exp\left(-O\left(\frac{NH_1}{d}\right)\right), \tag{7}$$

*where* $H_1 = \min_{i \in \{2, ..., K\}} \frac{\Delta_{(i)}}{i}$.

*Proof Sketch.* Let $p_e^{\mathcal{A}}(\mathcal{M}, U)$ denote the error probability for algorithm $\mathcal{A}$ to solve the problem 5.4, with specific unitary matrix $U$ and POVM $\mathcal{M}$. The error probability of $\mathcal{A}$ to solve the problem 5.4 satisfying $e_N^{\mathcal{A}} \ge \int_{\mathcal{M} \in \mathcal{D}_{\mathcal{M}}} \int_{U \sim \text{Haar}} p_e^{\mathcal{A}}(\mathcal{M}, U) \mathbb{1}\{U \in \mathbb{U}_{\mathcal{M}}\} d\mathcal{M} dU$.

When the $i$-th quantum state is measured using $\mathcal{M}$, the measurement result follows a Bernoulli distribution with parameter $\mathrm{Tr}(M\rho_U)$. From Lemma 5.5 and the definition of $c(\mathcal{M}, U)$, we have $\mathrm{Tr}(M\rho_i|U) = c(\mathcal{M}, U)\alpha_i + \frac{\mathrm{Tr}(M)}{d}$.

If $c(\mathcal{M}, U) > 0$, we need to find the Bernoulli distribution with the largest parameter where the parameter of the $i$-th Bernoulli distribution is $\mathrm{Tr}(M\rho_i|U)$, and we have $\mathrm{Tr}(M\rho_i|U) - \mathrm{Tr}(M\rho_j|U) = c(\mathcal{M}, U)\left[\sqrt{\frac{dz_i-1}{d-1}} - \sqrt{\frac{dz_j-1}{d-1}}\right]$.

Since $\mathrm{Tr}(M) > 16$ and according to the definition of $\mathbb{U}(\mathcal{M})$ and $M$, for $U \in \mathbb{U}$ we have $\mathrm{Tr}(M\rho_i|U) \in \left[\frac{\mathrm{Tr}(M)}{2d}, 1 - \frac{\mathrm{Tr}(M)}{2d}\right]$, and $1 - \frac{\mathrm{Tr}(M)}{2d} \geq \frac{1}{2}$. Then, according to the problem of best arm identification problem with Bernoulli distribution, we can demonstrate that $p_e^{\mathcal{A}}(U, \mathcal{M}) \geq \exp\left(-O\left(\frac{NH_1}{d}\right)\right)$. Then we have $e_N^{\mathcal{A}} \geq \exp\left(-O\left(\frac{NH_1}{d}\right)\right)$.

For any algorithm $\mathcal{A}_{\mathcal{D}}$ addressing Problem 5.4 cannot identify the purest random quantum state with an error probability lower than $\exp\left(-O\left(\frac{NH_1}{d}\right)\right)$. According to Lemma 5.3, we can complete the proof. The proof details are provided in Appendix C.3. $\qquad\square$

# 6 PQSI with coherent measurement

In this section, we investigate the problem of purest quantum state identification with coherent measurement and propose an algorithm to solve the purest quantum state identification with coherent measurement based on the SWAP test.

The SWAP test is a quantum algorithm designed to assess the similarity between two quantum states. It offers a method for estimating these states' fidelity to quantify their closeness. We use the SWAP test in Figure 1 to estimate the purity of the quantum state $\rho$ in the unknown quantum state set $S$. The measurement results in Figure 1 have a probability of $\frac{1+\mathrm{Tr}(\rho^2)}{2}$ for 0 and a probability of $\frac{1-\mathrm{Tr}(\rho^2)}{2}$ for 1. The details of the algorithm are shown in Algorithm 2.

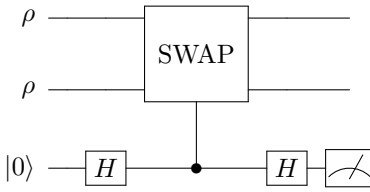

Figure 1: The SWAP test circuit.

---

**Algorithm 2** Coherent measurement based algorithm for solving PQSI problem (CM-PQSI)

---

   **Input:** Copy access to $S = \{\rho_1, \ldots, \rho_K\}$, sample number $N$.
   **Initialization:** Set $S_0 = \{\rho_1, \ldots, \rho_K\}$, $\overline{\log}(K) = \frac{1}{2} + \sum_{i=2}^{K} \frac{1}{i}$, $N_0 = 0$ and $N_k = \left\lceil \frac{1}{2\overline{\log}(K)} \frac{N-K}{K+1-k} \right\rceil$, for $k \in \{1, \ldots, K-1\}$.
   **for** $i = 1, \ldots, K-1$ **do**
      For all $\sigma \in S_{i-1}$, use SWAP test as Figure 1 for $N_k - N_{k-1}$ rounds, and set the outputs as $x_{(\sigma, N_{k-1}+1)}, \ldots, x_{(\sigma, N_k)}$.
      For all $\sigma \in S_{i-1}$, let $w(\sigma, k) = \frac{1}{N_k} \sum_{i=1}^{N_k} x_{(\sigma, i)}$.
      Let $S_k = S_{k-1} \setminus \arg\min_{\rho \in S_{k-1}} w(\rho, k)$.
   **end for**
   Output the quantum state $\rho'$ in $S_{k-1}$.

---

**Theorem 6.1.** *The probability of error of Algorithm 2 satisfies*

$$e_N \leq \frac{K(K-1)}{2} \exp\left(-\frac{NH_2}{8\overline{\log}(K)}\right), \tag{8}$$

*where $H_2 = \min_{i \in \{2,\dots,K\}} \frac{\Delta_{(i)}^2}{i}$.*

*Proof Sketch.* The outputs of the SWAP test are within the range $[0, 1]$ and are independent. Thus, we can apply the Hoeffding inequality to complete the proof. Detailed explanations of the proof are included in Appendix D. □

To further clarify the role of coherent measurements, we compare the theoretical performance of the proposed algorithms both with and without the SWAP test. The essential difference lies in the scaling behavior of the error probability bounds across different purity-gap regimes.

When the purity gaps satisfy $\Delta_{(i)} > 1/d^2$, the lower bound for the algorithm **without** the SWAP test is

$$e_N \leq \exp\left(-\Omega\left(\frac{NH_1}{\log K \cdot 2^n}\right)\right), \text{ where } H_1 = \min_{i \in \{2,\dots,K\}} \frac{\Delta_{(i)}}{i}. \tag{9}$$

In contrast, the algorithm **with** the SWAP test (i.e., using coherent measurements) achieves

$$e_N \leq \exp\left(-\Omega\left(\frac{NH_2}{\log K}\right)\right), \text{ where } H_2 = \min_{i \in \{2,\dots,K\}} \frac{\Delta_{(i)}^2}{i}, \tag{10}$$

which eliminates the additional $2^n$ factor in the denominator. This implies that the coherent-measurement-based algorithm can attain the same error probability with only $O(1/2^n)$ of the sample complexity, thereby demonstrating a substantial advantage in this regime.

When the purity gaps are very small ($\Delta_{(i)} \leq 1/d^2$), both algorithms yield similar lower bounds of the form

$$e_N \leq \exp\left(-\Omega\left(\frac{NH_2}{\log K}\right)\right), \tag{11}$$

and thus exhibit comparable asymptotic performance. Nevertheless, even in this regime, the algorithm employing the SWAP test tends to achieve a smaller constant factor in the exponent, resulting in empirically faster convergence and lower error rates for finite sample sizes. Therefore, the coherent measurement strategy remains effective in practice, providing improvements in both asymptotic and finite-sample efficiency.

In summary, the advantage of the SWAP test becomes particularly evident when the purity differences between quantum states are not vanishingly small. In most realistic scenarios, such tiny gaps can be regarded as negligible, and the corresponding states can be considered effectively equivalent. Consequently, for practical tasks that aim to distinguish states with noticeable purity differences, the coherent measurement strategy via the SWAP test offers a clear improvement in sample efficiency. This observation also highlights the essential role of quantum memory in enabling coherent measurements.

## 7  Conclusion and Outlook

In this study, we propose a pivotal problem in quantum testing, termed purest quantum state identification (PQSI). This framework applies to various quantum computing and quantum communication tasks. We develop two distinct algorithms to address this problem under different settings. When the learner utilizes incoherent (single-copy) measurement, the upper bound on the error probability of our algorithm is given by $\exp\left(-\Omega\left(\frac{NH_1}{\log(K)2^n}\right)\right)$. When the learner is allowed to use coherent (two-copy) measurement, the upper bound on the error probability is given by $\exp\left(-\Omega\left(\frac{NH_2}{\log(K)}\right)\right)$. By examining the error probabilities of these two algorithms, we can discern the advantage of the coherent measurement over the incoherent one. Furthermore, we establish that for any algorithm utilizing a randomly fixed incoherent two-outcome POVM to solve the PQSI, its error probability is lower bounded by $\exp\left(-O\left(\frac{NH_1}{2^n}\right)\right)$. Our results lay the groundwork for further investigations into the best quantum state identification. We aim to establish a lower bound for PQSI problems across all POVM bases in future work. Several open questions remain to be addressed, including identifying the nearest quantum state with minimal trace distance and achieving the best quantum state identification with fixed confidence.

## 8   Acknowledgement

This work was partially supported by Innovation Program for Quantum Science and Technology (Grant No. 2021ZD0302901), National Natural Science Foundation of China (Grant No. 62102388).

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

Table 1: Description of commonly used-notations

| Notation | Description |
|---|---|
| $n$ | the qubit number of the quantum state |
| $d$ | $d = 2^n$ |
| $S$ | the set of the unknown quantum state |
| $K$ | $K = |S|$ |
| $\rho_i$ | the $i$-th quantum state in $S$ |
| $\rho_{(i)}$ | the $i$-th purest quantum state in $S$ |
| $\rho^\star = \rho_{i\star}$ | the purest quantum state in $S$ |
| $\Delta_i$ | $\Delta_i = \text{Tr}(\rho_{i\star}^2) - \text{Tr}(\rho_i^2)$ |
| $\Delta_{(i)}$ | $\Delta_{(i)} = \text{Tr}(\rho_{i\star}^2) - \text{Tr}(\rho_{(i)}^2)$ |
| $\{|i\rangle\langle i|\}_{i=0}^{d-1}$ | a fixed orthogonal basis in $\mathbb{C}^{d \times d}$ |
| $I_d$ | $d$-dimensional identity matrix |
| $\mathbb{U}(d)$ | the set of $d \times d$ unitary matrix |

# A   Auxiliary tools

## A.1   Description of commonly used-notations

## A.2   Probability inequalities for sums of bounded random variables

In this paper, we utilize the following inequalities, which are provided for the sake of completeness.

**Theorem A.1** (Chebyshev's Inequality)**.** *Let $X$ be any random variable with expected value $\mu = \mathbb{E}[X]$ and finite variance $\text{Var}(X)$. Then, for any real number $\varepsilon > 0$:*

$$\mathbb{P}(|X - \mu| \geq \varepsilon) \leq \frac{\text{Var}(X)}{\varepsilon^2}. \tag{12}$$

**Theorem A.2** (Hoeffding's Inequality)**.** *If $X_1, X_2, ..., X_n$ are independent with $\mathbb{P}(a \leq X_i \leq b) = 1$ and common mean $\mu$ then for any $\varepsilon > 0$*

$$\mathbb{P}\left(\left|\frac{1}{n}\sum_{i=1}^{n} X_i - \mu\right| > \varepsilon\right) \leq 2\exp\left(\frac{-2n\varepsilon^2}{(b-a)^2}\right). \tag{13}$$

**Theorem A.3** (Bernstein's Inequality)**.** *If $X_1, ..., X_n$ are independent bounded random variables such that $\mathbb{E}[X_i] = 0$ for all $i \in \{1, ..., n\}$ and $\mathbb{P}(|X_i| \leq c) = 1$ then, for any $\epsilon > 0$,*

$$\mathbb{P}\left(\left|\frac{1}{n}\sum_{i=1}^{n} X_i\right| \geq \varepsilon\right) \leq \exp\left(-\frac{n\varepsilon^2}{2\sigma^2 + 2c\epsilon/3}\right), \tag{14}$$

*where $\sigma^2 = \frac{1}{n}\sum_{i=1}^{n} \text{Var}(X_i)$.*

## A.3   Properties of Haar unitary matrix

For a locally compact topological group, its Haar measure is a unique nonzero left-invariant measure (or right-invariant, depending on the formulation) under group operations. The Haar unitary matrix is the Haar measure on the unitary matrix group and is the concept of drawing unitary matrices uniformly at random. The formal definition of Haar unitary matrix is as follows:

**Definition A.4.** The Haar unitary matrix is the unique probability measure $\mu_H$ that is both left and right invariant over the unitary matrix group, i.e., for all integrable functions $f$ and for all unitary matrix $V$, we have:

$$\int_{U \sim \text{Haar}} f(U)dU = \int_{U \sim \text{Haar}} f(UV)dU = \int_{U \sim \text{Haar}} f(VU)dU. \tag{15}$$

For any unit column vector $\boldsymbol{x} \in \mathbb{C}^d$, we have

$$\mathbb{E}_{U \sim \text{Haar}}[f(U\boldsymbol{x})] = \mathbb{E}_{\psi \sim \mathbb{C}^d}[f(|\psi\rangle)]. \tag{16}$$

We will use the following lemma to complete our proofs in this paper.

**Lemma A.5** (see Lemma 22 of Ref.[27]). *Let $A, B, C$ be Hermitian matrices. Then*

$$\mathbb{E}_{\psi \sim \mathbb{C}^d} \langle \psi | A | \psi \rangle = \frac{1}{d} \text{Tr}(A), \tag{17}$$

*and*

$$\mathbb{E}_{\psi \sim \mathbb{C}^d} \langle \psi | A | \psi \rangle \langle \psi | B | \psi \rangle = \frac{1}{d(d+1)} (\text{Tr}(A)\text{Tr}(B) + \text{Tr}(AB)), \tag{18}$$

*and*

$$\mathbb{E}_{\psi \sim \mathbb{C}^d} \langle \psi | A | \psi \rangle \langle \psi | B | \psi \rangle \langle \psi | C | \psi \rangle = \frac{1}{d(d+1)(d+2)} (\text{Tr}(A)\text{Tr}(B)\text{Tr}(C) + \text{Tr}(AB)\text{Tr}(C) \\ + \text{Tr}(A)\text{Tr}(BC) + \text{Tr}(CA)\text{Tr}(B) + \text{Tr}(ABC)), \tag{19}$$

*and*

$$\mathbb{E}_{\psi \sim \mathbb{C}^d} \langle \psi | A | \psi \rangle \langle \psi | B | \psi \rangle \langle \psi | C | \psi \rangle \langle \psi | D | \psi \rangle = \frac{1}{d(d+1)(d+2)(d+3)} \sum_{\pi \in S_4} \text{Tr}(A \otimes B \otimes C \otimes D \cdot P_d(\pi)), \tag{20}$$

*where $P_d(\pi) = \sum_{i_1,...,i_k=0}^{d-1} |i_{\pi^{-1}(1)}, ..., i_{\pi^{-1}(4)}\rangle \langle i_1, ..., i_k|$ and $\pi$ is permutation on 4 items.*

# B  Proof of IMPQSI with incoherent measurement

## B.1  Proof of property for purity collision

**Lemma B.1.** *The expectation and variance of the purity estimation satisfying*

$$\mathbb{E}[\tilde{g}] = \frac{m-1}{m} \sum_{i=0}^{d-1} p_i^2, \tag{21}$$

*and*

$$\text{Var}[\tilde{g}] \leq \frac{2\mathbb{E}[\tilde{g}]}{m^2} + \frac{4}{m} \sum_{i=0}^{d-1} p_i^3. \tag{22}$$

*Proof.* The expectation of $\tilde{g}$ satisfying

$$\begin{aligned}
\mathbb{E}[\tilde{g}] &= \mathbb{E}\left[ \frac{1}{m^2} \sum_{i=0}^{d-1} \left[ \sum_{j=1}^{m} \mathbb{1}\{x_j = i\} \right]^2 \right] - \frac{1}{m} \\
&= \mathbb{E}\left[ \frac{1}{m^2} \sum_{i=0}^{d-1} \left[ \sum_{j=1}^{m} \sum_{k=1}^{m} \mathbb{1}\{x_j = i\}\mathbb{1}\{x_k = i\} \right] \right] - \frac{1}{m} \\
&= \mathbb{E}\left[ \frac{1}{m^2} \sum_{i=0}^{d-1} \left[ \sum_{j=1}^{m} \sum_{k=1}^{m} \mathbb{1}\{x_j = i\}\mathbb{1}\{x_k = i\} \right] \right] - \frac{1}{m} \\
&= \mathbb{E}\left[ \frac{1}{m^2} \sum_{i=0}^{d-1} \left[ \sum_{j=1}^{m} \mathbb{1}\{x_j = i\} + \sum_{j=1}^{m} \sum_{k \neq j} \mathbb{1}\{x_j = i\}\mathbb{1}\{x_k = i\} \right] \right] - \frac{1}{m} \\
&= \frac{1}{m^2} \sum_{i=0}^{d-1} \sum_{j=1}^{m} \mathbb{E}\left[\mathbb{1}\{x_j = i\}\right] + \frac{1}{m^2} \sum_{j=1}^{m} \sum_{k \neq j} \mathbb{E}[\mathbb{1}\{x_j = i\}\mathbb{1}\{x_k = i\}] - \frac{1}{m} \\
&= \frac{m}{m^2} + \frac{m-1}{m} \sum_{j=1}^{m} p_j^2 - \frac{1}{m} = \frac{m-1}{m} \sum_{j=1}^{m} p_j^2.
\end{aligned} \tag{23}$$

The expectation of $\tilde{g}^2$ satisfying

$$
\begin{aligned}
\mathbb{E}[\tilde{g}^2] =& \mathbb{E}\left[\left[\left[\frac{1}{m^2}\sum_{i=0}^{d-1}\left[\sum_{j=1}^{m}\mathbb{1}\{x_j = i\}\right]^2 - \frac{1}{m}\right]\right]^2\right] \\
=& \frac{1}{m^4}\mathbb{E}\left[\sum_{j_1 \neq j_2, l_1 \neq l_2}^{m}\left[\sum_{i,k}^{d-1}\mathbb{1}\{x_{j_1} = i\}\mathbb{1}\{x_{j_2} = i\}\mathbb{1}\{x_{l_1} = k\}\mathbb{1}\{x_{l_2} = k\}\right]\right] \\
=& \frac{1}{m^4}\left[2m(m-1)\mathbb{E}[\tilde{g}] + m(m-1)(m-2)(m-3)\mathbb{E}[\tilde{g}]^2 + 4m(m-1)(m-2)\sum_{j=1}^{n}p_j^3\right] \\
\leq& \frac{2}{m^2}\mathbb{E}[\tilde{g}] + \mathbb{E}[\tilde{g}]^2 + \frac{4}{m}\sum_{i=0}^{d-1}p_i^3
\end{aligned}
$$

Then, we have

$$
\begin{aligned}
\mathrm{Var}[\tilde{g}] &= \mathbb{E}[\tilde{g}^2] - \mathbb{E}[\tilde{g}]^2 \\
&\leq \frac{1}{m^2}\mathbb{E}[\tilde{g}] + \frac{2}{m}\sum_{i=0}^{d-1}p_i^3.
\end{aligned}
\tag{24}
$$

$\square$

### B.2 proof of Theorem 4.1

By using the techniques similar to [27], we can prove the following lemma:

**Lemma B.2** (See Lemma 16 of [27])**.** *The expectation of $w(\rho, k)$ and $\tilde{g}_{\rho,j}$ in Algorithm 1 satisfies*

$$
\mathbb{E}[w(\rho, k)] = \mathbb{E}[\tilde{g}_{\rho,j}] = \frac{(m-1)(1+\mathrm{Tr}(\rho^2))}{m(d+1)},
\tag{25}
$$

*and the variance of $\tilde{g}(\rho, j)$ satisfies*

$$
\mathrm{Var}(\tilde{g}(\rho, j)) = O\left(\frac{1}{d^3} + \frac{1}{m^2 d} + \frac{1}{md^2}\right).
\tag{26}
$$

By the definition of $w(\cdot, \cdot)$ and the definition of $\Delta_{(\cdot)}$, we have

$$
\begin{aligned}
&\mathbb{P}(w(\rho^\star, k) \leq w(\rho_{(i)}, k)) \\
=& \mathbb{P}\left((w(\rho_{(i)}, k) - w(\rho^\star, k)) \geq \frac{(m-1)\Delta_{(i)}}{m}\right).
\end{aligned}
$$

Since $w(\cdot, \cdot) \in [0, 1]$, by Lemma B.2 and Bernstein's inequality, we have

$$
\begin{aligned}
&\mathbb{P}\left((w(\rho_{(i)}, k) - w(\rho^\star, k)) \geq \frac{(m-1)\Delta_{(i)}}{m}\right) \\
\leq& \exp\left(-\frac{\lfloor\frac{N_k}{m}\rfloor\left(\frac{m-1}{m(d+1)}\Delta_{(i)}\right)^2}{O(\frac{1}{d^3} + \frac{1}{m^2 d} + \frac{1}{md^2}) + \frac{2\Delta_{(i)}}{3(d+1)}}\right) \\
\leq& \exp\left(-\Omega\left(\frac{\sqrt{c}N_k\Delta_{(i)}}{d}\right)\right).
\end{aligned}
\tag{27}
$$

By a union bound of error probability, we have

$$e_n \leq \sum_{k=1}^{K-1} \sum_{i=K+1-k}^{K} \mathbb{P}(w(\rho^\star, k) \leq w(\rho_{(i)}, n_k))$$

$$\leq \sum_{k=1}^{K-1} \sum_{i=K+1-k}^{K} \exp\left(-\Omega\left(\frac{\sqrt{c}N_k \Delta_{(i)}}{d}\right)\right) \tag{28}$$

$$\leq \sum_{k=1}^{K-1} k \exp\left(-\Omega\left(\frac{\sqrt{c}N_k \Delta_{(K+1-k)}}{d}\right)\right).$$

By definition of $N_k$, we have

$$\frac{\sqrt{c}N_k \Delta_{(K+1-k)}}{d}$$

$$= \left\lceil \frac{\sqrt{c}}{\log(K)} \frac{N-K}{K+1-k} \right\rceil \frac{\Delta_{(K+1-k)}}{d} \tag{29}$$

$$= \Theta\left(\frac{\sqrt{c}N}{\log(K)} \times \frac{\Delta_{(K+1-k)}}{K+1-k}\right).$$

Combining equation (28) and (29), we have

$$e_n \leq \sum_{k=1}^{K-1} k \exp\left(-\Omega\left(\frac{\sqrt{c}N}{\log(K)d} \times \frac{\Delta_{(K+1-k)}}{K+1-k}\right)\right)$$

$$\leq \frac{K(K-1)}{2} \exp\left(-\Omega\left(\frac{\sqrt{c}NH_1}{\log(K)d}\right)\right)$$

where $H_1 = \min_{i \in \{1,\ldots,K\}} \frac{\Delta_{(i)}}{i}$.

### B.3   proof of Lemma 4.2

*Proof.* By the definition of $w(\cdot, \cdot)$ and the definition of $\Delta_{(\cdot)}$, we have

$$\mathbb{P}(w(\rho^\star, k) \leq w(\rho_{(i)}, k))$$

$$= \mathbb{P}\left((w(\rho_{(i)}, k) - w(\rho^\star, k)) \geq \frac{(m-1)\Delta_{(i)}}{m}\right).$$

Since $w(\cdot, \cdot) \in [0, 1]$, by Lemma B.2 and Bernstein's inequality, we have

$$\mathbb{P}\left((w(\rho_{(i)}, k) - w(\rho^\star, k)) \geq \frac{(m-1)\Delta_{(i)}}{m}\right)$$

$$\leq \exp\left(-\frac{\lfloor \frac{N_k}{d} \rfloor \left(\frac{m-1}{m(d+1)}\Delta_{(i)}\right)^2}{O(\frac{1}{d^3}) + \frac{2\Delta_{(i)}}{3(d+1)}}\right) \tag{30}$$

$$\leq \exp\left(-\Omega\left(\min\left(\frac{N_k \Delta_{(i)}}{d^2}, N_k \Delta_{(i)}^2\right)\right)\right).$$

By a union bound of error probability, we have

$$e_n \leq \sum_{k=1}^{K-1} \sum_{i=K+1-k}^{K} \mathbb{P}(w(\rho^\star, k) \leq w(\rho_{(i)}, n_k))$$

$$\leq \sum_{k=1}^{K-1} \sum_{i=K+1-k}^{K} \exp\left(-\Omega\left(\min\left(\frac{N_k \Delta_{(i)}}{d^2}, N_k \Delta_{(i)}^2\right)\right)\right) \tag{31}$$

$$\leq \sum_{k=1}^{K-1} k \exp\left(-\Omega\left(\min\left(\frac{N_k \Delta_{(K+1-k)}}{d^2}, N_k \Delta_{(K+1-k)}^2\right)\right)\right).$$

By definition of $N_k$, and combining equation (31), we have

$$e_N \leq \frac{K(K-1)}{2} \exp\left(-\Omega\left(\min\left(\frac{NH_2}{\overline{\log(K)}}, \frac{NH_1}{\overline{\log(K)d^2}}\right)\right)\right),$$

where $H_1 = \min_{i \in \{2,...,K\}} \frac{\Delta_{(i)}}{i}$, and $H_2 = \min_{i \in \{2,...,K\}} \frac{\Delta_{(i)}^2}{i}$.

$\square$

## C  Proof of lower bound

### C.1  Proof of Lemma 5.3

Suppose that there exists such an algorithm $\mathcal{A}$ satisfying that the error probability of $\mathcal{A}$ for solving Problem 1.1 with the quantum state set $S_\rho$ whose error probability is less than $e_N$. Let $p_{S_\rho}(x_1, y_1; ...; x_N, y_N; z)$ denote the probability of the event satisfying

1. for $i \in 1, ..., N$, in the round $N$, the algorithm $\mathcal{A}$ select the $x_i$-th quantum state for measurement, and its output is $y_i$;
2. the algorithm $A$ output $z$-th quantum state at the end.

Furthermore, let $q_{S_\rho}(x_1, y_1; ...; x_N, y_N; z)$ denote the error probability corresponding to $p_{S_\rho}(x_1, y_1; ...; x_N, y_N; z)$.

Since when using $\mathcal{A}$ to solve the Problem 1.1, error probability is less than $e_N$. Then for all quantum state set $S = \{\rho_1, ..., \rho_K\}$, we have

$$\int_{(x_1, y_1,...,x_N, y_N, z)} q(x_1, y_1; ...; x_N, y_N; z) dp(x_1, y_1; ...; x_N, y_N; z) \leq e_N. \tag{32}$$

When the learner use the algorithm $\mathcal{A}$ to solve the problem 5.2, its error probability satifying

$$\begin{aligned}
e_N^{\mathcal{D}} &= \int_{x \sim \mathcal{D}'} \int_{(x_1, y_1,...,x_N, y_N, z)} q_{S_{\mathcal{D}(x)}}(x_1, y_1; ...; x_N, y_N; z) dp_{S_{\mathcal{D}(x)}}(x_1, y_1; ...; x_N, y_N; z) dx \\
&\leq \int_{x \sim \mathcal{D}'} e_N d_x \\
&\leq e_N.
\end{aligned} \tag{33}$$

then we can prove that if there is an algorithm $\mathcal{A}$ can solve the Problem 5.2 with the error probability less than $e_N$, then it can solve the Problem 1.1 with the error probability less than $e_N$. Furthermore, we can establish the proof by considering the contrapositive of this statement.

### C.2  proof of Lemma 5.5

Without loss of generality, assume that $M_0 = \arg\min_{M' \in \{M_0, M_1\}} \text{Tr}(M')$. According to the definition of POVM, there exists a unitary matrix $V$ and a diagonal matrix $\Sigma_0 = \text{diag}(b_0, ..., b_{d-1})$, where $b_0, ..., b_{d-1} \in [0, 1]$ such that

$$M_0 = V\Sigma_0 V^\dagger = \sum_{i=0}^{d-1} b_i V|i\rangle\langle i|V^\dagger,$$

$$M_1 = I - V\Sigma_0 V^\dagger = \sum_{i=0}^{d-1} (1 - b_i)V|i\rangle\langle i|V^\dagger.$$

We have

$$\text{Tr}(M_0) = \text{Tr}(\Sigma_0) = \sum_{i=0}^{d-1} b_i,$$

$$\text{Tr}(M_1) = \text{Tr}(I - \Sigma_0) = d - \sum_{i=0}^{d-1} b_i. \tag{34}$$

Let $p_{\mathcal{M}}(M|U)$ denote the probability that $M$ "accepts" the quantum state $\alpha U|0\rangle\langle 0|U^{\dagger} + \frac{1-\alpha}{d-1}I_d$. According to the property of the Haar measure and the identity matrix $I_d$, we have

$$
\begin{aligned}
&\mathbb{E}_{U\sim\text{Haar}}\left[p_{\mathcal{M}}^2(M_0|U)\right] \\
=&\mathbb{E}_{U\sim\text{Haar}}\left[\text{Tr}^2\left(M_0\left(\alpha U|0\rangle\langle 0|U^{\dagger} + \frac{1-\alpha}{d}I_d\right)\right)\right] \\
=&\mathbb{E}_{U\sim\text{Haar}}\left[\text{Tr}^2\left(\sum_{i=0}^{d-1}b_i V|i\rangle\langle i|V^{\dagger}\left(\alpha U|0\rangle\langle 0|U^{\dagger} + \frac{1-\alpha}{d}I_d\right)\right)\right] \\
=&\mathbb{E}_{U\sim\text{Haar}}\left[\left(\sum_{i=0}^{d-1}b_i\langle i|V^{\dagger}\left(\alpha U|0\rangle\langle 0|U^{\dagger} + \frac{1-\alpha}{d}I_d)\right)V|i\rangle\right)^2\right] \\
=&\mathbb{E}_{U\sim\text{Haar}}\left[\left(\sum_{i=0}^{d-1}\left(\alpha b_i\langle i|V^{\dagger}U|0\rangle\langle 0|U^{\dagger}V|i\rangle\right) + \sum_{i=0}^{d-1}b_i\frac{1-\alpha}{d}\langle i|V^{\dagger}I_dV|i\rangle\right)^2\right] \\
=&\mathbb{E}_{U\sim\text{Haar}}\left[\left(\sum_{i=0}^{d-1}\left(\alpha b_i\langle i|U|0\rangle\langle 0|U^{\dagger}|i\rangle\right) + \frac{1-\alpha}{d}\text{Tr}(M_0)\right)^2\right]
\end{aligned}
$$ (35)

Let $V_i$ is the matrix satisfying that $V_i|i\rangle = |0\rangle$, then $V_i$ is an unitary matrix and $V_i^{-1} = V_i^{\dagger}$, we have

$$
\begin{aligned}
&\mathbb{E}_{U\sim\text{Haar}}\left[p_{\mathcal{M}}^2(M_0|U)\right] \\
=&\mathbb{E}_{U\sim\text{Haar}}\left[\left(\sum_{i=0}^{d-1}\left(\alpha b_i\langle i|U|0\rangle\langle 0|U^{\dagger}|i\rangle\right) + \frac{1-\alpha}{d}\text{Tr}(M_0)\right)^2\right] \\
=&\mathbb{E}_{U\sim\text{Haar}}\left[\left(\sum_{i=0}^{d-1}\left(\alpha b_i\langle 0|U|i\rangle\langle i|U^{\dagger}|0\rangle\right) + \frac{1-\alpha}{d}\text{Tr}(M_0)\right)^2\right] \\
=&\mathbb{E}_{\psi\sim\mathbb{C}^d}\left[\left(\sum_{i=0}^{d-1}\left(\alpha b_i\langle\psi|i\rangle\langle i|\psi\rangle\right) + \frac{1-\alpha}{d}\text{Tr}(M_0)\right)^2\right] \\
=&\mathbb{E}_{\psi\sim\mathbb{C}^d}\left[\alpha^2\sum_{i=0}^{d-1}b_i^2\langle\psi|i\rangle\langle i|\psi\rangle\langle\psi|i\rangle\langle i|\psi\rangle + \alpha^2\sum_{i=0}^{d-1}\sum_{j\neq i}b_ib_j\langle\psi|i\rangle\langle i|\psi\rangle\langle\psi|j\rangle\langle j|\psi\rangle \right. \\
&\left. +2\sum_{i=0}^{d-1}\left(\alpha b_i\langle\psi|i\rangle\langle i|\psi\rangle\right)\frac{1-\alpha}{d}\text{Tr}(M_0) + \left(\frac{1-\alpha}{d}\right)^2\text{Tr}^2(M_0)\right]
\end{aligned}
$$ (36)

According to the Lemma A.5, we have for $i,j \in \{0,...,d-1\}, i\neq j$,

$$
\mathbb{E}_{\psi\sim\mathbb{C}^d}\langle\psi|i\rangle\langle i|\psi\rangle = \frac{1}{d},
$$ (37)

and

$$
\mathbb{E}_{\psi\sim\mathbb{C}^d}\langle\psi|i\rangle\langle i|\psi\rangle\langle\psi|i\rangle\langle i|\psi\rangle = \frac{2}{d(d+1)},
$$ (38)

and similarly

$$
\mathbb{E}_{\psi\sim\mathbb{C}^d}\langle\psi|i\rangle\langle i|\psi\rangle\langle\psi|j\rangle\langle j|\psi\rangle = \frac{1}{d(d+1)}.
$$ (39)

According to Equation (36), (37),(38) and (39), we have

$$\mathbb{E}_{U \sim \text{Haar}} \left[ p_{\mathcal{M}}^2(M_0|U) \right]$$

$$= \frac{2\alpha^2}{d(d+1)} \sum_{i=0}^{d-1} b_i^2 + \frac{\alpha^2}{d(d+1)} \sum_{i=0}^{d-1} \sum_{j \neq i} b_i b_j + \frac{2\alpha(1-\alpha)}{d^2} \text{Tr}(M_0) \sum_{i=0}^{d-1} b_i + \left( \frac{1-\alpha}{d} \right)^2 \text{Tr}^2(M_0)$$

$$= \frac{\alpha^2}{d(d+1)} \sum_{i=0}^{d-1} b_i^2 + \frac{\alpha^2}{d(d+1)} \left( \sum_{i=0}^{d-1} b_i \right)^2 + \frac{2\alpha(1-\alpha)}{d^2} \text{Tr}(M_0) \sum_{i=0}^{d-1} b_i + \left( \frac{1-\alpha}{d} \right)^2 \text{Tr}^2(M_0)$$

$$= \frac{\alpha^2}{d(d+1)} \sum_{i=0}^{d-1} b_i^2 + \frac{\alpha^2}{d(d+1)} \text{Tr}^2(M_0) + \frac{2\alpha(1-\alpha)}{d^2} \text{Tr}^2(M_0) + \left( \frac{1-\alpha}{d} \right)^2 \text{Tr}^2(M_0)$$

$$= \frac{\alpha^2}{d(d+1)} \sum_{i=0}^{d-1} b_i^2 + \left[ \frac{1}{d^2} - \frac{\alpha^2}{d^2(d+1)} \right] \text{Tr}^2(M_0),$$

(40)

and

$$\mathbb{E}_{U \sim \text{Haar}} \left[ p_{\mathcal{M}}(M_0|U) \right]$$

$$= \mathbb{E}_{U \sim \text{Haar}} \left[ \text{Tr} \left( M_0 \left( \alpha U|0\rangle\langle 0|U^\dagger + \frac{1-\alpha}{d} I_d \right) \right) \right]$$

$$= \mathbb{E}_{U \sim \text{Haar}} \left[ \text{Tr} \left( \sum_{i=0}^{d-1} b_i V|i\rangle\langle i|V^\dagger \left( \alpha U|0\rangle\langle 0|U^\dagger + \frac{1-\alpha}{d} I_d \right) \right) \right]$$

(41)

$$= \mathbb{E}_{U \sim \text{Haar}} \left[ \text{Tr} \left( \sum_{i=0}^{d-1} b_i V|i\rangle\langle i|V^\dagger \left( \alpha U|0\rangle\langle 0|U^\dagger + \frac{1-\alpha}{d} I_d \right) \right) \right] = \frac{\text{Tr}(M_0)}{d}.$$

Then the variance of $p_{\mathcal{M}}(M_0|U)$ is given by

$$\text{Var} \left[ p_{\mathcal{M}}(M_0|U) \right]$$

$$= \mathbb{E}_{U \sim \text{Haar}} \left[ p_{\mathcal{M}}^2(M_0|U) \right] - \left( \mathbb{E}_{U \sim \text{Haar}} \left[ p_{\mathcal{M}}(M_0|U) \right] \right)^2$$

$$= \frac{\alpha^2}{d(d+1)} \sum_{i=0}^{d-1} b_i^2 + \left[ \frac{1}{d^2} - \frac{\alpha^2}{d^2(d+1)} \right] \text{Tr}^2(M_0) - \frac{\text{Tr}^2(M_0)}{d^2}$$

$$= \frac{\alpha^2}{d(d+1)} \sum_{i=0}^{d-1} b_i^2 - \frac{\alpha^2}{d^2(d+1)} \text{Tr}^2(M_0)$$

$$= \frac{\alpha^2}{d^2(d+1)} \left[ d \sum_{i=0}^{d-1} b_i^2 - \text{Tr}^2(M_0) \right]$$

(42)

$$\leq \frac{\alpha^2}{d^2(d+1)} \left[ d \sum_{i=0}^{d-1} b_i - \text{Tr}^2(M_0) \right]$$

$$= \frac{\alpha^2}{d^2(d+1)} \left[ d\text{Tr}(M_0) - \text{Tr}^2(M_0) \right]$$

$$\leq \frac{\alpha^2}{d^2(d+1)} \left[ d\text{Tr}(M_0) - \text{Tr}^2(M_0) \right]$$

$$\leq \frac{\alpha^2 \text{Tr}(M_0)}{d(d+1)}.$$

From Chebyshev's Inequality, we have

$$\mathbb{P}_{U \sim \text{Haar}} \left[ \left| p_{\mathcal{M}}(M_0|U) - \frac{\text{Tr}(M_0)}{d} \right| \geq \frac{2\alpha\sqrt{\text{Tr}(M_0)}}{d} \right] < \frac{1}{4}.$$

(43)

And

$$p_{\mathcal{M}}(M_0|U)$$

$$=\mathrm{Tr}\left(\sum_{i=0}^{d-1}b_iV|i\rangle\langle i|V^\dagger\left(\alpha U|0\rangle\langle 0|U^\dagger+\frac{1-\alpha}{d}I_d\right)\right)$$

$$=\mathrm{Tr}\left(\sum_{i=0}^{d-1}b_iV|i\rangle\langle i|V^\dagger\left(\alpha U|0\rangle\langle 0|U^\dagger+\frac{\alpha}{d}I_d\right)\right)+\mathrm{Tr}\left(\sum_{i=0}^{d-1}b_iV|i\rangle\langle i|V^\dagger\left(\frac{1}{d}I_d\right)\right) \tag{44}$$

$$=\alpha\mathrm{Tr}\left(\sum_{i=0}^{d-1}b_iV|i\rangle\langle i|V^\dagger\left(U|0\rangle\langle 0|U^\dagger+\frac{1}{d}I_d\right)\right)+\frac{M_0}{d}.$$

Let $c(\mathcal{M},U)=\mathrm{Tr}\left(\sum_{i=0}^{d-1}b_iV|i\rangle\langle i|V^\dagger\left(U|0\rangle\langle 0|U^\dagger+\frac{1}{d}I_d\right)\right)$, we have

$$p_{\mathcal{M}}(M_0|U)-\frac{M_0}{d}=c(\mathcal{M},U)\alpha. \tag{45}$$

### C.3 Proof of Theorem 5.6

We will use the following theorem to complete the proof:

**Theorem C.1** (see Theorem 4 of Ref. [32])**.** *Let $\nu_1,...,\nu_K$ be Bernoulli distributions with parameters in $[a,1-a]$, $a\in(0,1/2)$. For any forecaster, there exists a permutation $\sigma:\{1,...,K\}\to\{1,...,K\}$ such that the probability error of the forecaster on the bandit problem defined by $\tilde{\nu}_1=\nu_{\sigma(1)},...,\tilde{\nu}_K=\nu_{\sigma(K)}$ satisfies*

$$e_n\geq\exp\left(-\frac{(5+o(1))nH}{pa(1-a)}\right), \tag{46}$$

*where $H=\min_i\frac{(\mathbb{E}[\nu^*]-\mathbb{E}[\nu_{(i)}])^2}{i}$.*

Let $p_e^{\mathcal{A}}(\mathcal{M},U)$ denote the error probability for algorithm $\mathcal{A}$ to solve the problem 5.4, with specific unitary matrix $U$ and POVM $\mathcal{M}$, and $M=\min_{M'\in\{M_0,M_1\}}\mathrm{Tr}(M')$. Then the error probability of $\mathcal{A}$ to solve the problem 5.4 satisfying

$$e_N^{\mathcal{A}}=\int_{\mathcal{M}\in\mathcal{D}_{\mathcal{M}}}\int_{U\sim\mathrm{Haar}}p_e^{\mathcal{A}}(\mathcal{M},U)d\mathcal{M}dU$$

$$\geq\int_{\mathcal{M}\in\mathcal{D}_{\mathcal{M}}}\int_{U\sim\mathrm{Haar}}p_e^{\mathcal{A}}(\mathcal{M},U)\mathbb{1}\{U\in\mathbb{U}_{\mathcal{M}}\}d\mathcal{M}dU. \tag{47}$$

The first line corresponds to the deifintion of $e_N^{\mathcal{A}}$, the second line corresponds to that $p_e^{\mathcal{A}}(\mathcal{M},U)\geq 0$ and $\mathbb{U}_{\mathcal{M}}$ is a subset of unitary matrix.

When the $i$-th quantum state is measured using $\mathcal{M}$, the process in which the output result is accepted by $M$ follows a Bernoulli distribution with parameter $\mathrm{Tr}(M\rho_U)$. From Lemma 5.5 and the definition of $c(\mathcal{M},U)$, we have

$$\mathrm{Tr}(M\rho_i|U)=c(U,\mathcal{M})\alpha_i+\frac{\mathrm{Tr}(M)}{d}. \tag{48}$$

If $c(\mathcal{M},U)>0$, we need to find the Bernoulli distribution with the largest parameter where the parameter of the $i$-th Bernoulli distribution is $\mathrm{Tr}(M\rho_i|U)=c(\mathcal{M},U)\alpha_i+\frac{\mathrm{Tr}(M)}{d}$. Then we have

$$\mathrm{Tr}(M\rho_i|U)-\mathrm{Tr}(M\rho_j|U)$$

$$=c(\mathcal{M},U)\alpha_i-c(\mathcal{M},U)\alpha_j$$

$$=c(\mathcal{M},U)\left[\sqrt{\frac{dz_i-1}{d-1}}-\sqrt{\frac{dz_j-1}{d-1}}\right]. \tag{49}$$

Since $\mathrm{Tr}(M)>16$, for $U\in\mathbb{U}(\mathcal{M})$ we have

$$\mathrm{Tr}(M\rho_i|U)=c(\mathcal{M},U)\alpha_i+\frac{\mathrm{Tr}(M)}{d}$$

$$\geq-\frac{2\sqrt{\mathrm{Tr}(M)}}{d}+\frac{Tr(M)}{d}\geq\frac{\mathrm{Tr}(M)}{2d}. \tag{50}$$

And since $M = \arg\min_{M' \in \{M_0, M-1\}} \mathrm{Tr}(M')$, we have $\mathrm{Tr}(M) \leq \frac{1}{2}$, then for $U \in \mathbb{U}$ we have

$$\mathrm{Tr}(M\rho_i|U) \in \left[\frac{\mathrm{Tr}(M)}{2d}, 1 - \frac{\mathrm{Tr}(M)}{2d}\right], \tag{51}$$

and

$$1 - \frac{\mathrm{Tr}(M)}{2d} \geq \frac{1}{2}. \tag{52}$$

According to Theorem C.1 and the definition of $\mathbb{U}_{\mathcal{M}}$, for $U \in \mathbb{U}_M$ we have

$$
\begin{aligned}
p_e^{\mathcal{A}}(\mathcal{M}, U) &\geq \exp\left(-O\left(\frac{N}{\Omega\left(\frac{\mathrm{Tr}(M)}{d}\right)\left(1 - \Omega\left(\frac{\mathrm{Tr}(M)}{d}\right)\right)} \min_i \frac{\left(\mathrm{Tr}(M\rho^\star|U) - \mathrm{Tr}(M\rho_{(i)}|U)\right)^2}{i}\right)\right) \\
&= \exp\left(-O\left(\frac{Nc^2(\mathcal{M}, U)}{\Omega\left(\frac{\mathrm{Tr}(M)}{d}\right)} \min_i \frac{\left(\sqrt{dz_{i^\star} - 1} - \sqrt{dz_{(i)} - 1}\right)^2}{i(d-1)}\right)\right) \\
&= \exp\left(-O\left(\frac{Nc^2(\mathcal{M}, U)}{\Omega\left(\frac{\mathrm{Tr}(M)}{d}\right)} \min_i \frac{dz_{i^\star} - 1 + dz_{(i)} - 1 - 2\sqrt{(dz_{i^\star} - 1)(dz_{(i)} - 1)}}{i(d-1)}\right)\right) \\
&\geq \exp\left(-O\left(\frac{Nc^2(\mathcal{M}, U)}{\Omega\left(\frac{\mathrm{Tr}(M)}{d}\right)} \min_i \frac{[dz_{i^\star} - 1] - [dz_{(i)} - 1]}{i(d-1)}\right)\right) \\
&= \exp\left(-O\left(\frac{Nc^2(\mathcal{M}, U)}{\Omega\left(\frac{\mathrm{Tr}(M)}{d}\right)} \min_i \frac{d\Delta_{(i)}}{i(d-1)}\right)\right)
\end{aligned}
\tag{53}
$$

According to the definition of $\mathbb{U}_{\mathcal{M}}$, for $U \in \mathbb{U}_M$ we have

$$c(\mathcal{M}, U) \in \left(-\frac{2\sqrt{\mathrm{Tr}(M)}}{d}, \frac{2\sqrt{\mathrm{Tr}(M)}}{d}\right). \tag{54}$$

According to Equation (53) and Equation (54), we have

$$
\begin{aligned}
p_e^{\mathcal{A}}(U, \mathcal{M}) &\geq \exp\left(-O\left(\frac{Nc^2(U, \mathcal{M})}{\Omega\left(\frac{\mathrm{Tr}(M)}{d}\right)} \min_i \frac{d\Delta_{(i)}}{i(d-1)}\right)\right) \\
&\geq \exp\left(-O\left(\frac{N\left(\frac{\sqrt{\mathrm{Tr}(M)}}{d}\right)^2}{\Omega\left(\frac{\mathrm{Tr}(M)}{d}\right)} \min_i \frac{d\Delta_{(i)}}{i(d-1)}\right)\right) \\
&\geq \exp\left(-O\left(\frac{N}{d} \min_i \frac{\Delta_{(i)}}{i}\right)\right) \\
&= \exp\left(-O\left(\frac{NH_1}{d}\right)\right).
\end{aligned}
\tag{55}
$$

Similarly, if $c(U, \mathcal{M}) <= 0$, we have

$$p_e^{\mathcal{A}}(U, \mathcal{M}) \geq \exp\left(-O\left(\frac{NH_1}{d}\right)\right). \tag{56}$$

According to Equation (47), (55), (56) we have

$$
\begin{aligned}
e_N^{\mathcal{A}} &\geq \int_{\mathcal{M} \in \mathcal{D}_{\mathcal{M}}} \int_{U \sim \text{Haar}} p_e^{\mathcal{A}}(\mathcal{M}, U) \mathbb{1}\{U \in \mathbb{U}_{\mathcal{M}}\} d\mathcal{M} dU \\
&\geq \int_{\mathcal{M} \in \mathcal{D}_{\mathcal{M}}} \int_{U \sim \text{Haar}} \exp\left(-O\left(\frac{NH_1}{d}\right)\right) \mathbb{1}\{U \in \mathbb{U}_{\mathcal{M}}\} d\mathcal{M} dU \\
&\geq \exp\left(-O\left(\frac{NH_1}{d}\right)\right) \int_{U \sim \text{Haar}} \mathbb{1}\{U \in \mathbb{U}_{\mathcal{M}}\} d\mathcal{M} dU \\
&\geq \exp\left(-O\left(\frac{NH_1}{d}\right)\right) \int_{\mathcal{M} \in \mathcal{D}_{\mathcal{M}}} \frac{3}{4} d\mathcal{M} \\
&\geq \exp\left(-O\left(\frac{NH_1}{d}\right)\right).
\end{aligned}
$$

Then for any algorithm $\mathcal{A}_{\mathcal{D}}$ addressing Problem 5.4 cannot identify the purest random quantum state with an error probability lower than $\exp\left(-O\left(\frac{NH_1}{d}\right)\right)$. According to Lemma 5.3, we can complete the proof.

## D Proof of Theorem 6.1

By the definition of $w(\cdot, \cdot)$ and the definition of $\Delta_{(\cdot)}$, we have

$$
\begin{aligned}
&\mathbb{P}(w(\rho^\star, k) \leq w(\rho_{(i)}, k)) \\
&= \mathbb{P}\left((w(\rho_{(i)}, k) - w(\rho^\star, k)) \geq \frac{\Delta_{(i)}}{2}\right).
\end{aligned}
$$

Since $x_{(\cdot, \cdot)} \in [0, 1]$ and Hoeffding's inequality, we have

$$
\begin{aligned}
&\mathbb{P}\left((w(\rho_{(i)}, k) - w(\rho^\star, k)) \geq \frac{\Delta_{(i)}}{2}\right) \\
&\leq \exp\left(-\frac{N_k}{2}\left(\frac{\Delta_{(i)}}{2}\right)^2\right) \\
&\leq \exp\left(-\frac{N_k \Delta_{(i)}^2}{8}\right).
\end{aligned}
\tag{57}
$$

By a union bound of error probability, we have

$$
\begin{aligned}
e_n &\leq \sum_{k=1}^{K-1} \sum_{i=K+1-k}^{K} \mathbb{P}(w(\rho^\star, k) \leq w(\rho_{(i)}, n_k)) \\
&\leq \sum_{k=1}^{K-1} \sum_{i=K+1-k}^{K} \exp\left(-\frac{N_k \Delta_{(K+1-k)}^2}{8}\right) \\
&\leq \sum_{k=1}^{K-1} k \exp\left(-\frac{N_k \Delta_{(K+1-k)}^2}{8}\right).
\end{aligned}
\tag{58}
$$

By definition of $N_k$, we have

$$
N_k \Delta_{(K+1-k)}^2 = \left\lceil \frac{1}{\overline{\log}(K)} \frac{N-K}{K+1-k} \right\rceil \Delta_{(K+1-k)}^2 \leq \frac{N-K}{\overline{\log}(K)} \times \frac{\Delta_{(K+1-k)}^2}{K+1-k}.
\tag{59}
$$

Combining equation (58) and (59), we have

$$
\begin{aligned}
e_n &\leq \sum_{k=1}^{K-1} k \exp\left(-\frac{N-K}{8\overline{\log}(K)} \times \frac{\Delta_{K+1-k}^2}{K+1-k}\right) \\
&\leq \frac{K(K-1)}{2} \exp\left(-\frac{NH_2}{8\overline{\log}(K)}\right),
\end{aligned}
$$

where $H_2 = \min_{i \in \{1,\dots,K\}} \frac{\Delta_{(i)}^2}{i}$.

# E  Simulation Results

To empirically validate our theoretical results, we performed numerical simulations on classical simulators to evaluate the performance of the proposed method under finite-sample conditions. We considered a collection of 10 randomly generated 6-qubit quantum states of the form

$$\rho_i = (1 - \lambda_i) |\psi_i\rangle\langle\psi_i| + \lambda_i \frac{I_d}{d}, \tag{60}$$

where each $|\psi_i\rangle$ is a pure state sampled uniformly at random, and the mixing parameter $\lambda_i$ is chosen such that the purity $\text{Tr}(\rho_i^2) = 0.5 + 0.04i$. The task is to identify the state with the highest purity given $N = 30{,}000$ copies of the unknown quantum states.

We tested the following three algorithms:

- **IM_PQSI:** our proposed adaptive algorithm using only incoherent measurements.
- **CM_PQSI:** a variant incorporating coherent measurements via the SWAP test.
- **Unadaptive:** a non-adaptive baseline that uniformly allocates samples and uses the same purity estimator as IM_PQSI.

Each algorithm was run for 100 independent trials. The results are summarized as follows:

- **IM_PQSI** achieves a $53\%$ success rate, with lower average purity 0.8736 and higher variance 0.001319.
- **CM_PQSI** achieves perfect success in all 100 runs and consistently selects the state with purity 0.9.
- **Unadaptive** achieves a $43\%$ success rate, with lower average purity 0.86192 and higher variance 0.002145.

These results demonstrate that coherent measurements (via the SWAP test) substantially improve accuracy, confirming the theoretical prediction of their advantage. Moreover, comparing IM_PQSI with the Unadaptive algorithm shows that adaptivity enhances performance by allocating more samples to promising candidates, leading to higher estimation accuracy and success rates.

Due to the computational cost of high-dimensional quantum-state simulation, our experiments are restricted to small systems ($n = 6$) and moderate sample budgets ($N = 3 \times 10^4$). Nevertheless, the observed behavior aligns well with our theoretical predictions, and we expect the advantages of adaptive and coherent measurement strategies to become even more pronounced for larger-scale systems and higher sample budgets.

