# OpenReview forum: "Purest Quantum State Identification"
_NeurIPS.cc/2025/Conference — NeurIPS 2025 poster_

### Official Review · Reviewer_ATtA · 2025-06-26

**Clarity:** 3
**Significance:** 3
**Originality:** 3
**Rating:** 5
**Confidence:** 3

**Summary:**

This paper introduces the problem of Purest Quantum State Identification (PQSI). Given $K$ unknown $n$-qubit states, the goal is to identify the quantum state with the higest purity using $N$ total copies of these states. The authors propose algorithms for both incoherent (single-copy) and coherent (multi-copy) measurement strategies and provide upper bounds on error probability. They also establish a lower bound for the incoherent case.

**Questions:**

1. Have you considered the fixed-confidence setting? Could your algorithms be adapted to this setting rather than the fixed-budget framework?
2. Have you considered the lower bound for coherent measurements? How does it differ from the incoherent case, and what challenges arise in establishing it?
3. Is it possible to use alternative purity estimation methods such as quantum tomography? If so, how would this affect the performance of your algorithms?

**Ethical Concerns:**

["NO or VERY MINOR ethics concerns only"]

**Final Justification:**

While several theoretical aspects remain open and require further investigation, I believe this paper makes a meaningful step forward in addressing an important problem. Therefore, I lean toward acceptance and would like to maintain my positive score.

**Limitations:**

yes

**Quality:**

3

**Strengths And Weaknesses:**

Strengths:
1. The formulation of the PQSI problem is novel and opens a promising direction for integrating quantum computing with online learning. While both components (purity estimation in quantum computing and best-arm identification in online learning) are well studied individually, their combination is still non-trivial and requires careful algorithmic and theoretical design.
2. The paper provides rigorous theoretical results, including upper bounds for both incoherent and coherent measurement strategies, and a lower bound using carefully constructed hardness reductions.
3. Compared with existing studies on quantum online learning, this paper adopts a more natural problem formulation that does not rely on strong quantum oracles.

Weaknesses:
1. The paper lacks empirical or simulation-based validation to support the theoretical claims.
2. The presentation lacks key intuitions. For example, the collision estimator is conceptually simple, but its physical meaning and intuitions are not explained when presenting the algorithms.

Overall, I recommend accepting this paper. Integrating quantum computing and online learning is a timely and important topic, but existing studies usually rely on strong assumptions about quantum oracles, and the feedback models lack concrete counterparts in real-world quantum systems. In contrast, this paper proposes a more practical problem formulation.

---

> ### Author Rebuttal · Authors · 2025-07-30
>
> We sincerely appreciate your valuable time, effort, and constructive feedback in reviewing this paper. We are also grateful for your recognition of our work. For the potential concerns you bring up, we provide point-by-point responses in the following paragraphs.
>
> ---
>
> **Response to empirical or simulation-based validation(Weakness 1):**
>
> We sincerely thank the reviewer for this excellent suggestion. We agree that both numerical simulations and real experiments can significantly enhance the credibility and practical relevance of our theoretical results.
>
> However, implementing such experiments comes with notable challenges. On the simulation side, numerically modeling large-scale quantum systems requires repeated high-dimensional matrix operations, which is computationally intensive and limits us to relatively small system sizes. On the experimental side, realizing real quantum hardware experiments is currently constrained by the limited availability and accessibility of physical quantum devices.
>
> We fully recognize the importance of the reviewer’s suggestion. To this end, we conducted **numerical simulations** using classical simulators to evaluate the practical performance of our method in a finite-sample regime as follows.
>
> In our simulations, we considered a set of 10 randomly generated 6-qubit quantum states of the form
> $$
> \rho_i = (1 - \lambda_i) \frac{I_d}{d} + \lambda_i |\psi_i\rangle\langle\psi_i|,
> $$
> where $|\psi_i\rangle$ are pure states sampled uniformly at random, and the mixing parameter $\lambda_i$ is chosen so that the purity of $\rho_i$ is exactly $0.5 + 0.04i$ for $i = 1, \dots, 10$. The task is to identify the state with the highest purity using 30,000 copies of the unknown quantum states.
>
> We tested the following four algorithms:
>
> - **IM_PQSI**: Our proposed adaptive algorithm using only incoherent measurements.
> - **CM_PQSI**: A variant of our algorithm that incorporates coherent measurements (via the SWAP test).
> - **Unadaptive**: A non-adaptive baseline that uniformly allocates samples and uses the same purity estimator as IM_PQSI.
>
> Each algorithm was run for 100 independent trials. The results are as follows:
>
> - **CM_PQSI** achieves perfect success in all 100 runs and consistently selects the state with purity 0.9.
> - **IM_PQSI** achieves a 53% success rate, with lower average purity 0.8736 and higher variance 0.001319.
> - **Unadaptive** achieves a 43% success rate, with lower average purity 0.86192 and higher variance 0.002145.
>
> From these simulations, we find that the SWAP test (coherent measurement) significantly improves accuracy, demonstrating the advantage of coherent measurements. Comparing IM_PQSI with the Unadaptive algorithm shows that the adaptive strategy improves accuracy by allocating more samples to better-performing states, resulting in more accurate purity estimation and higher success rates.
>
> Due to the limitations of classical simulation resources, we are currently restricted to small-scale experiments, with both the system dimension $ d = 2^n $ and the total sample size $ N $ constrained.  However, based on our theoretical results, we expect that the advantage of our algorithm will become even more pronounced as both the quantum system dimension and sample budget increase.
>
> Our current simulations are relatively simple. We plan to perform more extensive simulations and real-device experiments in the future, and will include these results in the paper to further enhance the credibility of our conclusions.
>
> ---
>
> **Response to key intuitions(Weakness 2):**
>
> We agree with the reviewer that providing more intuitive explanations would improve the clarity of our presentation. In the revised version, we will enhance the exposition of key intuitions behind our algorithm design.
>
> In particular, regarding the use of the collision estimator, our rationale is as follows:
>
> The POVMs we adopt has $d$ possible outcomes. Let $p_i$ denote the probability of observing the $i$-th outcome. We show that the purity of a quantum state is closely related to $\sum_{i=1}^d p_i^2$. Therefore, an essential step in our algorithm is to estimate this quantity accurately.
>
> The collision estimator is a well-established technique in classical statistics for estimating the second moment $\sum_{i=1}^d p_i^2$. Motivated by this, we adopt the collision estimator as a natural and efficient tool for estimating quantum state purity in our setting.
>
> ---
>
> **Response to the fixed-confidence setting(Question 1):**
>
> We sincerely thank the reviewer for raising this insightful and important question. Indeed, we have considered the fixed-confidence setting as a potential extension of our framework. While we believe that our algorithm can be adapted to this setting with some modifications, conducting a satisfactory theoretical analysis within this framework remains a challenging task.
>
> The main difficulties are summarized as follows:
>
> * Modeling Assumptions: In classical best-arm identification under the fixed-confidence setting, it is typically assumed—to enable theoretical analysis—that all arms (distributions) belong to a one-parameter exponential family. In our setting, applying this assumption would imply that each quantum state in the candidate set is characterized by only a single scalar parameter, which is generally not realistic.
>
> * KL Divergence Analysis: Algorithm design and lower bound analysis in the fixed-confidence setting strongly rely on precise control of the Kullback–Leibler (KL) divergence between distributions. However, for Haar-random unitary measurements, the KL divergence between measurement outcome distributions has not been adequately studied in existing work. Most known results require that one of the compared quantum states be either a pure state or a maximally mixed state—conditions that do not hold in our general setting.
>
> * Alternative Approaches: We have also explored using alternative random matrix ensembles such as the Gaussian Orthogonal Ensemble (GOE) in place of Haar unitaries. Unfortunately, these substitutes introduce their own challenges and have not resolved the core analytical difficulties.
>
> In summary, the fixed-confidence setting is a valuable and meaningful direction. Although we have explored it, we have not yet found an analysis that meets the level of completeness and clarity we are aiming for. We plan to continue working on this important extension in future work.
>
>
>
> ---
>
> **Response to lower bound for coherent measurements(Question 2):**
>
> We sincerely thank the reviewer for this insightful question. We have indeed considered the problem of establishing a lower bound for coherent measurements, and we believe that the lower bound in this setting could be significantly smaller than that for incoherent measurements.
>
> Analyzing the coherent case introduces additional layers of complexity. In each round, one not only has to decide which quantum states to measure and which measurement basis to use, but also how many states to jointly measure in a single coherent operation. This additional degree of freedom greatly complicates both the modeling and theoretical analysis.
>
> Our current work primarily focuses on the algorithm design and lower bound analysis under the incoherent measurement model. Nonetheless, we agree that studying lower bounds and algorithmic strategies under the coherent setting is both important and technically challenging. We view this as an important direction and plan to investigate it in future work.
>
> ---
>
> **Response to alternative purity estimation(Question 3):**
>
> We thank the reviewer for raising this interesting and relevant question.
>
> Indeed, one could consider applying quantum state tomography as an alternative approach for estimating the purity of the quantum states. However, doing so would likely reduce the accuracy of the algorithm in the fixed-budget setting.
>
> Quantum tomography aims to reconstruct the full density matrix of a quantum state, including many properties that are only weakly related to purity. As a result, it becomes more difficult to estimate purity accurately.
>
> From a theoretical perspective, achieving trace distance $\epsilon$ via quantum state tomography requires $\Theta(d^3/\epsilon^2)$ measurements using general POVMs [1]. In contrast, our purity estimation method only requires $O(\sqrt{d}/\epsilon^2)$ samples to achieve the same error. As the dimension $d$ increases, tomography-based methods require significantly more measurements, which leads to reduced accuracy under a fixed sampling budget.
>
> ---
>
> We sincerely thank you again for your insightful questions. We will incorporate further discussion of these points in the revised version of the paper. If there are any further questions or points that require clarification, we would be grateful for the opportunity to address them.
>
> ---
>
> **Reference:**
>
> [1] Chen S, Huang B, Li J, et al. When does adaptivity help for quantum state learning?[C]//2023 IEEE 64th Annual Symposium on Foundations of Computer Science (FOCS). IEEE, 2023: 391-404.

---

> > ### Comment · Reviewer_ATtA · 2025-08-03
> >
> > Thanks for the detailed response. I have read the authors' rebuttal and the comments from other reviewers.
> >
> > Overall, I am satisfied with the response. While several theoretical aspects remain open and require further investigation, I believe this paper makes a meaningful step forward in addressing an important problem. Therefore, I would like to maintain my positive score.
> >
> > I encourage the authors to incorporate their simulation results and the intuition behind the collision estimator in the revised version of the paper.

---

### Official Review · Reviewer_mGRh · 2025-07-01

**Clarity:** 2
**Significance:** 3
**Originality:** 3
**Rating:** 4
**Confidence:** 4

**Summary:**

The authors propose two algorithms for purest quantum state identification. The first algorithm requires first a random unitary transformation of the quantum state, followed by a POVM measurement. Purity can be estimated by the probability of identical outcomes, after which the highest is found. The second algorithm utilizes the SWAP test to directly compute the purity instead of estimation.

**Questions:**

1. The algorithm for incoherent measurements seems to share similiarities with the classical shadows method proposed in Ref. [26] with slight modifications in for 4-unitary designs, projective measurements to POVM, and from estimating all values to an fixed accuracy to a search problem. Regardless, the main protocol seems similiar, with the random transformation and subsequent channel inversion that takes the form of computing identical probablities in this algorithm.

2. Given that Lemma A.5 only lists results up to the 3rd power, does this imply that the protocol only require 3-designs instead of 4-designs? In this case, we would only require random Clifford circuits, which would be surprising.

**Ethical Concerns:**

["NO or VERY MINOR ethics concerns only"]

**Final Justification:**

The authors have addressed my previous concerns, and I retain my original evaluation.

**Limitations:**

Yes

**Quality:**

3

**Strengths And Weaknesses:**

__Strengths__
1. The paper proposes an interesting question for quantum information.

2. The paper provides a lower bound in addition to algorithm results, adding more context to the performance of the algorithm itself.

__Weakness__
1. Not sure if the scope of the paper is aligned with the call for papers, particularly the theory track, though I suppose optimization could work.

2. Algorithm and results seem simililar to the classical shadows method proposed in Ref. [26]. See further Question 1.

3. The proof of requiring unitary 4-designs is not provided in the manuscript, and Haar random unitaries are assumed to be provided in the proof. See further Question 2.

---

> ### Author Rebuttal · Authors · 2025-07-29
>
> We sincerely appreciate your valuable time and effort in reviewing this paper. We thank the thoughtful feedback you provided, which significantly improved the quality of this paper. For the potential concerns you bring up, we provide point-by-point responses in the following paragraphs.
>
> ---
>
> **Response to the scope of the paper(Weakness 1)**
>
> Our paper is fundamentally centered on **online learning theory**, a core and active area of research regularly featured at NeurIPS. Although our application involves quantum states, the main problem we study is a sequential estimation task with bandit feedback—where the learner adaptively selects measurements based on past observations. This feedback-driven structure is standard in online learning and aligns closely with the NeurIPS community’s interests. Because online learning theory is a part of learning theory, we believe our work is more suitable for the theory track than the optimization track.
>
> The quantum setting introduces additional challenges, such as exponentially large state spaces and restricted observability, making the learning task particularly complex and meaningful. These features do not isolate our work from machine learning; rather, they provide a high-dimensional framework to explore the limits of online learning under novel constraints.
>
> Our contribution thus extends classical online learning theory into the quantum regime in a way that advances both fields. This fits well with NeurIPS’s mission of supporting research that pushes learning theory into new, high-impact domains.
>
> NeurIPS and other top-tier venues have a history of accepting theoretical works involving quantum aspects, e.g., [1][2][3]. In particular, [1]—a highly cited NeurIPS paper—addresses quantum-state online learning and quantum bandit problems. Compared to [1], our model is more grounded in current experimental constraints and has broader relevance to near-term quantum technologies.
>
> We firmly believe that our contribution fits well within the scope of NeurIPS and will be of interest to its community.
>
> ---
>
> **Response to the technology of the paper(Question 1)**
>
> We are grateful to the reviewer for pointing out this connection. Using random procedures to build unbiased estimators for quantum properties is indeed a powerful and widely used approach.
>
> While our method shares some general ideas with Ref. [26]—such as using random unitaries and measurement inversion—we make several important extensions:
>
> - We give a more detailed theoretical analysis focused on the specific task of purity estimation.
> - Our algorithm is adapted to the purity testing problem, instead of aiming to estimate all observables equally well.
> - We also apply online learning methods to improve estimation accuracy, which is especially useful given the practical limits of today’s quantum hardware.
>
> We believe that our contributions—from problem setting to algorithm design and theoretical results—provide meaningful progress in quantum property testing. They also better match the capabilities and constraints of current NISQ-era quantum devices.
>
> ---
>
> **Response to the unitary 4-designs(Question 2)**
>
> Thank you for this valuable question—it reflects a deep understanding of the topic. Indeed, our algorithm **requires unitary 4-designs**.
>
> This is because, in our protocol, estimating the purity of quantum states involves squaring the measurement outcomes. The expectation of the purity estimator relies on the expression  $\mathbb{E}_{U \sim \text{Haar}}[U^{\otimes 2} (\cdot)U^{\dagger\otimes 2}],$
>
> while the variance analysis of the estimator requires $\mathbb{E}_{U \sim \text{Haar}}[U^{\otimes 4} (\cdot) U^{\dagger\otimes 4}]$.
>
> Although Lemma A.5 lists results only up to the 3rd power, the 4th power analysis is in Lemma B.2, where we directly cite relevant results from previous work. Therefore, the protocol still requires unitary 4-designs to guarantee the correctness of the theoretical bounds.
>
> According to the algorithm described in [4], such quantities can be efficiently estimated using unitary 4-designs.
>
> We fully agree with the reviewer that performing the estimation using only unitary 3-designs would greatly enhance the practicality of the algorithm. This is a very interesting question, and we intend to investigate it further in future work.
>
> ---
>
> We sincerely thank the reviewer again for these insightful questions. We will incorporate further discussion of these points in the revised version of the paper. If there are any further questions or points that require clarification, we would be grateful for the opportunity to address them.
>
> ---
>
> **Reference:**
>
> [1] Aaronson S, Chen X, Hazan E, et al. Online learning of quantum states[J]. Advances in neural information processing systems, 2018, 31.
>
> [2] Quek Y, Arunachalam S, Smolin J A. Private learning implies quantum stability[J]. Advances in Neural Information Processing Systems, 2021, 34: 20503-20515.
>
> [3] Yang F, Jiang J, Zhang J, et al. Revisiting online quantum state learning[C]//Proceedings of the AAAI Conference on Artificial Intelligence. 2020, 34(04): 6607-6614.
>
> [4] Ma F, Huang H Y. How to construct random unitaries[C]//Proceedings of the 57th Annual ACM Symposium on Theory of Computing. 2025: 806-809.

---

> ### Comment · Reviewer_mGRh · 2025-08-03
>
> Thank you to the authors for addressing the concerns. I have read the authors' rebuttal as well as the other reviews.
>
> I am happy with the authors' response and I hope the authors can perhaps add technical results of 4-designs to Lemma A.5 as well as writing the proof of Lemma B.2 to make things clearer.
>
> Overall, I find that the work in the paper is rigorous, solid, and interesting, but perhaps not as groundbreaking and surprising. I will keep my positive valuation unchanged to this paper.

---

### Official Review · Reviewer_j8kw · 2025-07-03

**Clarity:** 2
**Significance:** 2
**Originality:** 2
**Rating:** 4
**Confidence:** 4

**Summary:**

This paper introduces Purest Quantum State Identification (PQSI) - the task of identifying the purest state out of a set of finite states. The authors propose algorithms for both single-copy and multi-copy measurement strategies, along with a lower bound for the former. The motivation is to identify the quantum state least affected by noise, which is framed as identifying the state with the highest purity.

**Questions:**

- The distinction between properties for a fixed POVM and properties averaged over Haar random POVMs is sometimes mixed up by the authors. This can drastically affect the results and their proofs require careful attention to how expectations and variances are taken.
- How do the authors get the second equation of 552? This seems incorrect.
- In particular, there are many unjustified steps/approximations in proofs, especially in Lemma B.1 (the variance approximation) and Lemma 5.5 (factor of 2 in variance and justification for Haar integral application is questionable). I suggest the authors carefully redo their technical analyses.
- Notation could be more consistent.

Update: these questions and concerns have been addressed in the comments and I am satisfied with the answers.

**Ethical Concerns:**

["NO or VERY MINOR ethics concerns only"]

**Final Justification:**

As explained in the revised review, the concerns around technical soundness have been addressed. Considering this, as well as the very positive reviews by the other reviewers, I am happy to change my score from reject to borderline accept.

**Limitations:**

yes

**Quality:**

3

**Strengths And Weaknesses:**

Strengths:
- The paper contains algorithms for both single and multi-copy measurement ability.
- The authors keep practical considerations in mind.

Weaknesses:
1.  Before delving into technical flaws, I believe this work is simply not suited for NeurIPS. The content and research direction has virtually nothing to do with machine learning. This type of work is probably better suited for a quantum theory journal/conference like QIP/TQC.

2.  The derived bounds on the error probability are all extremely loose. Am I perhaps missing something? For example in Thm 1.3 and 1.4, if the gaps H_1 and H_2 are small, this gives an upper bound close to 1. These are two of the main results in a completely theory-based paper.

3.  There are many technical issues in the proofs and I question the validity of the technical soundness.

Update after rebuttal:
1.  Indeed, this work is suitable for the online learning theory category, but I still maintain that it is more appropriate for QIP/TQC.
2.  The authors clarified that loose bounds in small gap regimes are an illustration of the hardness of the task and pointed to other analogous settings in RL for which this occurs which marginally puts this concern at ease.
3. My concerns around technical issues have been resolved by the author's response.

---

> ### Author Rebuttal · Authors · 2025-07-28
>
> We sincerely appreciate your valuable time and effort in reviewing this paper. We would like to emphasize that we are confident in the correctness and significance of our main results.
>
> We sincerely apologize for any confusion caused by insufficient explanation or minor typographical errors in the presentation. In the revised version, we will provide detailed justifications for each step to improve clarity and readability. We provide point-by-point responses to your questions in the following paragraphs.
>
> ---
>
> **Response to Relevance to NeurIPS and ML Community(Weakness 1):**
>
> Our paper is fundamentally centered on **online learning theory**, a core and active area of research regularly featured at NeurIPS. Although our application involves quantum states, the main problem we study is a sequential estimation task with bandit feedback—where the learner adaptively selects measurements based on past observations. This feedback-driven structure is standard in online learning and aligns closely with the NeurIPS community’s interests.
>
> The quantum setting introduces additional challenges, such as exponentially large state spaces and restricted observability, making the learning task particularly complex and meaningful. These features do not isolate our work from machine learning; rather, they provide a high-dimensional framework to explore the limits of online learning under novel constraints.
>
> Our contribution thus extends classical online learning theory into the quantum regime in a way that advances both fields. This fits well with NeurIPS’s mission of supporting research that pushes learning theory into new, high-impact domains.
>
> NeurIPS and other top-tier venues have a history of accepting theoretical works involving quantum aspects, e.g., [1][2][3]. In particular, [1]—a highly cited NeurIPS paper—addresses quantum-state online learning and quantum bandit problems. Compared to [1], our model is more grounded in current experimental constraints and has broader relevance to near-term quantum technologies.
>
> We firmly believe that our contribution fits well within the scope of NeurIPS and will be of interest to its community.
>
> ---
>
> **Response to the derived bounds on the error probability(Weakness 2) :**
>
> We would like to clarify that the bounds in Theorems 1.3 and 1.4 are not overly loose. When the gaps $H_1$ and $H_2$ are small, this indicates the presence of a suboptimal quantum state whose purity is very close to that of the optimal one. In such cases, the problem itself becomes intrinsically difficult, and a large number of samples is required to distinguish between nearly optimal states—reflected in the scaling of $N$ in our bound. This behavior is expected and consistent with similar phenomena in statistical learning and bandit literature. For instance, classical results on best-arm identification also exhibit dependence on such gap parameters [4].
>
> As an analogy, in the classical stochastic multi-armed bandit setting[5], the instance-dependent regret is often bounded by:
> $$\mathbb{E}[\text{Regret}(T)] \leq \sum_{a: \mu(a) < \mu^*} \frac{\log T}{\Delta(a)},$$
>
> where $\Delta(a) = \mu^* - \mu(a)$ is the gap between the optimal arm and arm $a$. When the gap is small, the regret bound becomes large, reflecting the increased difficulty in distinguishing between near-optimal choices. Extreme cases where the gap $\Delta(a)$ approaches zero do not invalidate the meaningfulness of the bound in general settings. A similar phenomenon is captured by our bounds through the gap parameters $H_1$ and $ H_2 $.
>
> Hence, our result is theoretically justified and meaningful.
>
> ---
>
> **Response to the distinction between properties for a fixed POVM and properties averaged over Haar random POVMs (Question 1):**
>
> We would like to assure you that our analyses concerning fixed POVMs and Haar random POVMs are correct. In our work, Haar random POVMs are used to perform measurements on quantum states for unbiased estimation of purity. In the theoretical analysis, we carefully account for the variance arising from both the randomness of Haar sampling and the measurement outcomes associated with the fixed POVM drawn from the Haar distribution. **Considering both sources of randomness is essential for a correct and complete analysis.** This should not be viewed as a conflation of the two types of measurements, but rather as the proper structure of the variance decomposition in our setting.
>
> ---
>
> **Response to  the second equation of 552 (Question 2) :**
>
> We would like to clarify that **the second equation in line 552 is correct.** The presence of the $\Omega$ notation is optional in the first line of 552. From Theorem C.1, we obtain the first line of equation 552:
> $$
> p_e^\mathcal{A}(\mathcal{M},U) \geq \exp\left(-O\left( \frac{N}{\left(\frac{\operatorname{Tr}(M)}{d}\right)\left(1-\frac{\operatorname{Tr}(M)}{d}\right)} \min_i \frac{\left(\operatorname{Tr}(M\rho^\star|U) - \operatorname{Tr}(M\rho_{(i)}|U)\right)^2}{i} \right)\right).
> $$
>
> In Equation (44), we showed that $1 - \frac{\operatorname{Tr}(M)}{2d} \geq \frac{1}{2}$. This allows us to safely replace the factor $1 - \frac{\operatorname{Tr}(M)}{d}$ with $1/2$ in the bound without affecting the inequality, since the substitution only loosens the lower bound. Furthermore, this constant $1/2$ appears inside the $O(\cdot)$ term and therefore can be omitted.
>
> Lastly, in Equation (41), we provided an alternative expression for the difference $\operatorname{Tr}(M\rho_i|U) - \operatorname{Tr}(M\rho_j|U)$. Substituting that into the first line of 552 yields the second equation directly.
>
> Since the expression is inside the $O(\cdot)$ notation, using “=” rather than “$\geq$” does not affect the validity of the bound.
>
>
> We will clarify this derivation in the revised version to avoid further confusion.
>
> ---
>
> **Response to the question of Lemma B.1 and Lemma 5.5(Question 3):**
>
> (a) **Lemma B.1** :  Here we briefly explain the key idea behind bounding $ \operatorname{Var}[\tilde{g}] $. According to the definition of $ \tilde{g} $, the first line of Equation (479) follows directly. When expanding the expression $ \sum_{i=0}^{d-1} \left[ \sum_{j=1}^m \mathbf{1}(x_j = i) \right]^2 $, the squared terms $ [\mathbf{1}(x_j = i) ]^2 $ contribute exactly $ m $ ones due to indicator properties. Their sum yields a term that cancels with the $ 1/m $ factor, explaining the second line of Equation (479). All remaining index patterns fall into three categories, whose expectations correspond to $ \mathbb{E}[\tilde{g}] $, $ \sum_{i=0}^{d-1} p_i^3 $, and $ \mathbb{E}[\tilde{g}]^2 $, respectively. These categories correspond to index patterns with: (1) two identical indices among four, (2) one pair of repeated indices, and (3) all distinct indices. Their respective counts are $ \Theta(m^2) $, $ \Theta(m^3) $, and $ m^4 - \Theta(m^3) $. Based on this classification, we obtain the third line of Equation (479). By further simplification, the fourth line of Equation (479) follows. We can show that
> $ \mathbb{E}[\tilde{g}^2] \leq c_1 \frac{\mathbb{E}[\tilde{g}]}{m^2} + c_2 \frac{\sum_{i=0}^{d-1} p_i^3}{m} + \mathbb{E}[\tilde{g}]^2 $, and hence derive the upper bound $ \operatorname{Var}[\tilde{g}] \leq c_1 \frac{\mathbb{E}[\tilde{g}]}{m^2} + c_2 \frac{\sum_{i=0}^{d-1} p_i^3}{m} $, which will be used in the proof of Lemma 5.2.
>
>
> (b) **Lemma 5.5 factor of 2 in variance**: The factor of 2 is necessary in our setting because we aim to ensure that more than $\frac{3}{4}$ of the probability satisfies the desired property. By Chebyshev’s inequality, $\mathbb{P}(|X - \mu| \geq \varepsilon) \leq \frac{\mathrm{Var}(X)}{\varepsilon^2}$, we require $\varepsilon = 2\sqrt{\mathrm{Var}(X)}$ to guarantee that $\mathbb{P}(|X - \mu| \geq \varepsilon) \leq \frac{1}{4}$, which implies $\mathbb{P}(|X - \mu| < \varepsilon) \geq 3/4$. This directly supports our probabilistic guarantee, and thus justifies the factor of 2.
>
> (c) **Lemma 5.5 Haar integral application:** In the proof of Lemma 5.5, we apply several algebraic identities that do not involve Haar randomness to simplify and derive expressions. For the part involving Haar integrals, the step is justified by standard results[6]: we transform expressions of the form $\int_{U \sim \text{Haar}} \langle 0 | U (\cdot) U^\dagger | 0 \rangle dU$ into $\int_{\psi \sim \mathbb{C}^d} \langle \psi | (\cdot) | \psi \rangle dU$,  and similarly,  $\int_{U \sim \text{Haar}} f(V U) (\cdot) dU$ into  $\int_{U \sim \text{Haar}} f(U) (\cdot) dU$, where $V$ is a unitary matrix.  These are valid changes of variables due to the unitary invariance of the Haar measure. Therefore, the application of Haar integrals in our derivation is mathematically sound.
>
>
> We will revise the manuscript to clearly state the reasoning behind each step and improve the overall clarity and readability.
>
> ---
>
> **Response to the Notation(Question 4):**
>
> Thank you for your comment. We will revise the manuscript to ensure consistent and clear notation throughout.
>
> ---
>
> Once again, we sincerely thank you for your time and insightful feedback. We are confident in the correctness and significance of our results, and we will carefully revise the paper to enhance its clarity and readability in accordance with your suggestions.
>
> **If you have any further questions or concerns, please let us know—we would be glad to respond promptly.**
>
> ---
>
> Reference:
>
> [1] Aaronson S, Chen X, Hazan E, et al. Online learning of quantum states. NIPS 2018
>
> [2] Quek Y, Arunachalam S, Smolin J A. Private learning implies quantum stability. NIPS 2021
>
> [3] Yang F, Jiang J, Zhang J, et al. Revisiting online quantum state learning//AAAI 2020.
>
> [4] Audibert J Y, Bubeck S. Best arm identification in multi-armed bandits//COLT-23th.
>
> [5] Slivkins A. Introduction to multi-armed bandits[J]. Foundations and Trends in Machine Learning, 2019.
>
> [6] Mele A A. Introduction to Haar measure tools in quantum information: A beginner's tutorial. Quantum, 2024, 8: 1340.

---

> > ### Comment · Reviewer_j8kw · 2025-08-05
> >
> > I would like to thank the authors for their comprehensive response to my concerns and questions. In particular, the additional context for the bounds was helpful and would probably be useful as a footnote in the manuscript if not in the main text already. Regarding my concerns around the technical aspects of the manuscript, I believe these are adequately addressed by the authors. As such, I am happy to revise my score positively.

---

> ### Comment · Area_Chair_xAaj · 2025-08-05
> **respond to rebuttal**
>
> Dear Reviewer j8kw,
>
> Please read the authors' rebuttal and other reviewers' comments as early as possible! As requested by NeurIPS code of conduct, You have to provide your further feedback and engage in a discussion with the authors.
>
> AC

---

### Official Review · Reviewer_7uPd · 2025-07-03

**Clarity:** 2
**Significance:** 3
**Originality:** 3
**Rating:** 4
**Confidence:** 2

**Summary:**

This paper addresses the problem of quantum noise in the context of quantum system identification and introduces the task of purest quantum state identification. Theoretical analysis is provided to support the proposed approach.

**Questions:**

1）Regarding the lower bound, it would be helpful to clarify which specific factors or parameters determine the lower bound in the proposed analysis. For example, does it depend on the system dimension, noise level, or properties of the quantum state ensemble?
2）In Algorithm 2, an additional SWAP test circuit is introduced to perform coherent measurements. However, it remains unclear how this component contributes to improving the accuracy or robustness of purest quantum state identification. A more detailed discussion or theoretical justification would strengthen the argument.
3）Experimental validation of the proposed method is currently lacking. Incorporating numerical simulations or prototype experiments—such as using noisy intermediate-scale quantum (NISQ) devices or classical simulators—would significantly enhance the applicability of the approach.

**Ethical Concerns:**

["NO or VERY MINOR ethics concerns only"]

**Final Justification:**

Thank you for the author's response and for addressing my question. I would like to maintain my original  score.

**Limitations:**

Including simulation experiments would further strengthen the theoretical claims and improve the overall completeness of the work.

**Paper Formatting Concerns:**

no formatting concerns

**Quality:**

3

**Strengths And Weaknesses:**

Strengths:
The authors introduce an incoherent measurement algorithm to solve the purest quantum state identification problem and provide a theoretical analysis of the error probability and lower bound.
Weaknesses:
No simulation experiments are presented. Including such results would help validate the effectiveness of the proposed algorithm.

---

> ### Author Rebuttal · Authors · 2025-07-30
>
> We sincerely appreciate your valuable time and effort in reviewing this paper. We thank you for the thoughtful feedback you provided, which significantly improved the quality of this paper. For the potential concerns you bring up, we provide point-by-point responses in the following paragraphs.
>
> ---
>
> **Response to factors or parameters in lower bound(Question 1):**
>
> Thank you for your helpful comment. We agree that explaining the parameters behind our lower bound will make the result clearer to readers. Our lower bound is
> $$
> e_N \geq \exp\left(- O\left(\frac{N H_1}{d}\right)\right),
> $$
> where $H_1 = \min_{i \in \{2, \ldots, K\}} \frac{\Delta_{(i)}}{i}$ and $d = 2^n$ is the system dimension.
>
> We briefly explain how each factor affects this bound:
>
> - **Number of samples $N$:**
>   A larger $N$ leads to a smaller error lower bound.
>
> - **System dimension $d$:**
>   As the number of qubits increases, $d = 2^n$ grows exponentially. This makes the problem harder, since the required number of samples increases rapidly with the system size.
>
> - **Quantum state ensemble (through $H_1$):**
>   $H_1$ reflects the purity gap between the optimal state and the others. A smaller gap (i.e., states with similar purities) makes it more difficult to identify the optimal one, which matches the intuitive hardness of distinguishing nearly identical states.
>
> - **Noise level:**
>   Our model assumes noiseless measurements. If measurement noise is present, both the algorithm and the analysis would need to be adjusted based on the specific noise model. On the other hand, our method can be used to identify the system with the lowest noise level, since the noise of a quantum channel is often characterized by the purity of its Choi state.
>
> We thank the reviewer again for the suggestion and will include this explanation in the revision.
>
> ---
>
> **Response to the contribution of swap test(Question 2):**
>
> Thank you for the insightful comment. We agree that clarifying the role of the SWAP test in our algorithm is important for understanding its contribution to performance. Below we explain its theoretical advantage by comparing the error probability upper bounds with and without the SWAP test.
>
> When the SWAP test is used, our algorithm achieves the following upper bound:
> $$
> e_N \leq \exp\left(- \Omega\left(\frac{N H_2}{\log K}\right) \right),
> $$
> where $H_2 = \min_{i \in \{2, \ldots, K\}} \frac{\Delta_{(i)}^2}{i}$ and $\Delta_{(i)}$ denotes the purity gap between the $i$-th best state and the optimal one.
>
> We now consider two scenarios to compare this with the algorithm that does not use the SWAP test:
>
> 1. **When $\Delta_{(i)} > 1/d^2$**:
>    The lower bound for the algorithm *without* the SWAP test becomes $e_N \leq \exp\left(- \Omega\left(\frac{N H_1}{\log K \cdot 2^n}\right) \right)$,where $H_1 = \min_{i \in \{2, \ldots, K\}} \frac{\Delta_{(i)}}{i}$.  Comparing the two bounds, we see that the dominant difference lies in the $2^n$ factor. In this regime, the algorithm using coherent measurements (i.e., the SWAP test) can achieve the same error probability bound with $O(1/2^n)$ samples, demonstrating a substantial advantage.
>
> 2. **When $\Delta_{(i)} \leq 1/d^2$**:
>    In this case, both algorithms yield similar lower bounds of the form $e_N \leq \exp\left(- \Omega\left(\frac{N H_2}{\log K}\right) \right)$, and thus exhibit comparable performance.
>
> These comparisons show that the SWAP test significantly improves the sample efficiency when there is a not small purity gap between the optimal and suboptimal states. On the other hand, when the purity differences are extremely small ( $\leq 1/d^2$), the two algorithms perform similarly. However, in realistic scenarios, such small purity gaps can often be ignored, and the corresponding states can be treated as equally good choices. In typical applications, we aim to distinguish states with noticeable purity differences, where the benefit of coherent measurements via the SWAP test becomes clearly evident.
>
> We will incorporate this explanation in the revised version to better highlight the value of the SWAP test.
>
> ---
>
> **Response to experimental validation(Question 3):**
>
> We sincerely thank the reviewer for this excellent suggestion. We agree that both numerical simulations and real experiments can significantly enhance the credibility and practical relevance of our theoretical results.
>
> However, implementing such experiments comes with notable challenges. On the simulation side, numerically modeling large-scale quantum systems requires repeated high-dimensional matrix operations, which is computationally intensive and limits us to relatively small system sizes. On the experimental side, realizing real quantum hardware experiments is currently constrained by the limited availability and accessibility of physical quantum devices.
>
> We fully recognize the importance of the reviewer’s suggestion. To this end, we conducted **numerical simulations** using classical simulators to evaluate the practical performance of our method in a finite-sample regime as follows.
>
> In our simulations, we considered a set of 10 randomly generated 6-qubit quantum states of the form
> $$
> \rho_i = (1 - \lambda_i) \frac{I_d}{d} + \lambda_i |\psi_i\rangle\langle\psi_i|,
> $$
> where $|\psi_i\rangle$ are pure states sampled uniformly at random, and the mixing parameter $\lambda_i$ is chosen so that the purity of $\rho_i$ is exactly $0.5 + 0.04i$ for $i = 1, \dots, 10$. The task is to identify the state with the highest purity using 30,000 copies of the unknown quantum states.
>
> We tested the following four algorithms:
>
> - **IM_PQSI**: Our proposed adaptive algorithm using only incoherent measurements.
> - **CM_PQSI**: A variant of our algorithm that incorporates coherent measurements (via the SWAP test).
> - **Unadaptive**: A non-adaptive baseline that uniformly allocates samples and uses the same purity estimator as IM_PQSI.
>
> Each algorithm was run for 100 independent trials. The results are as follows:
>
> - **CM_PQSI** achieves perfect success in all 100 runs and consistently selects the state with purity 0.9.
> - **IM_PQSI** achieves a 53% success rate, with lower average purity 0.8736 and higher variance 0.001319.
> - **Unadaptive** achieves a 43% success rate, with lower average purity 0.86192 and higher variance 0.002145.
>
> From these simulations, we find that the SWAP test (coherent measurement) significantly improves accuracy, demonstrating the advantage of coherent measurements. Comparing IM_PQSI with the Unadaptive algorithm shows that the adaptive strategy improves accuracy by allocating more samples to better-performing states, resulting in more accurate purity estimation and higher success rates.
>
> Due to the limitations of classical simulation resources, we are currently restricted to small-scale experiments, with both the system dimension $ d = 2^n $ and the total sample size $N$ constrained.  However, based on our theoretical results, we expect that the advantage of our algorithm will become even more pronounced as both the quantum system dimension and sample budget increase.
>
> Our current simulations are relatively simple. We plan to perform more extensive simulations and real-device experiments in the future, and will include these results in the paper to further enhance the credibility of our conclusions.
>
> ---
>
> We sincerely thank the reviewer again for these insightful questions. We will incorporate further discussion of these points in the revised version of the paper. If there are any further questions or points that require clarification, we would be grateful for the opportunity to address them.

---

### Author Response · Authors · 2025-08-08
**Grateful for the reviewers’ and AC’s efforts**

As the reviewer–author discussion period comes to an end, we would like to sincerely thank all reviewers for their positive evaluation of our work and for the constructive, insightful feedback provided. We are also grateful to the Area Chair for the kind assistance during the review process. We will incorporate the reviewers’ suggestions in future revisions and further explore the related research questions. Once again, we truly appreciate the time and effort contributed by all reviewers and the Area Chair.

---

### Decision · Program_Chairs · 2025-09-17

**Decision:**

Accept (poster)

**Comment:**

This paper addresses the quantum noise in quantum system identification and presents the problem termed purest quantum state identification (PQSI). The proposed framework is applicable to various quantum computing and quantum communication tasks. Algorithms for incoherent (single-copy) and coherent (multi-copy) measurement strategies are proposed with theoretical results on the error probability. Reviewers find that the paper addresses an interesting problem with solid theoretical results and a more natural, practical formulation. After rebuttal, reviewers acknowledge their concerns resolved and favor the acceptance for this paper. The AC goes through all the comments and agrees with the reviewers. Accept is recommended for this paper.